

# SMLFire1.0: a stochastic machine learning (SML) model for wildfire activity in the western United States

Jatan Buch[1], A. Park Williams[2], Caroline Juang[1, 3], Winslow D. Hansen[4], and Pierre Gentine[5]

[1]Lamont-Doherty Earth Observatory, Columbia University, Palisades, NY, USA
[2]Department of Geography, University of California, Los Angeles, CA, USA
[3]Department of Earth and Environmental Sciences, Columbia University, New York, NY, USA
[4]Cary Institute of Ecosystem Studies, Millbrook, NY, USA
[5]Department of Earth and Environmental Engineering, Columbia University, New York, NY, USA

**Correspondence:** Jatan Buch (jb4625@columbia.edu)

**Abstract.** The annual area burned due to wildfires in the western United States (WUS) increased by more than 300% between 1984 and 2020. However, accounting for the nonlinear, spatially heterogeneous interactions between climate, vegetation, and human predictors driving the trends in fire frequency and sizes at different spatial scales remains a challenging problem for statistical fire models. Here we introduce a novel stochastic machine learning (SML) framework, SMLFire1.0, to model observed fire frequencies and sizes in $12\,\text{km} \times 12\,\text{km}$ grid cells across the WUS. This framework is implemented using Mixture Density Networks trained on a wide suite of input predictors. The modeled WUS fire frequency corresponds well with observations at both monthly ($r = 0.94$) and annual ($r = 0.85$) timescales, as do the monthly ($r = 0.90$) and annual ($r = 0.88$) area burned. Moreover, the annual time series of both fire variables exhibit strong correlations ($r \geq 0.6$) in 16 out of 18 ecoregions. Our ML model captures the interannual variability and the distinct multidecade increases in annual area burned for both forested and non-forested ecoregions. Evaluating predictor importance with Shapley additive explanations, we find that fire month vapor pressure deficit (VPD) is the dominant driver of fire frequencies and sizes across the WUS, followed by 1000-hour dead fuel moisture (FM1000), total monthly precipitation (Prec), mean daily maximum temperature (Tmax), and fraction of grassland cover in a grid cell. Our findings serve as a promising use case of ML techniques for wildfire prediction in particular and extreme event modeling more broadly. They also highlight the power of ML driven parameterizations for potential implementation in the fire modules of Dynamic Global Vegetation Models (DGVMs) and Earth System Models (ESMs).

## 1 Introduction

Wildfire is an important biophysical process that structures natural and anthropogenic systems, and is, in turn, affected by climate, vegetation, and humans (Bowman et al., 2009; Krawchuk et al., 2009). The relative strength of each driver and the interactions between them, however, vary across multiple spatial and temporal scales. For instance, sediment charcoal records indicate that while global biomass burning, a proxy for total area burned, responded strongly to warming and drought in the past, these relationships weakened beginning in the late 1800s in many regions due to changes in land use as well as more active fire management (Marlon et al., 2008). Modern satellite observations between 1998 and 2015 (Giglio et al., 2013), on the other





hand, indicate divergent trends along tree cover gradients (Andela et al., 2017); although the decreased fire activity in grasslands and shrublands contributed to the overall decline in global burned area, forest area burned increased across the globe (Zheng

et al., 2021). In fact, for regions like the western United States (WUS), there was a $\gtrsim 300\%$ increase in the total area burned between 1984 and 2020, promoted by high flammability of fuels induced by more frequent hot temperature extremes, rising atmospheric aridity, and prolonged drought-like conditions (Dennison et al., 2014; Abatzoglou and Williams, 2016; Zhuang et al., 2021; Kuhn-Régnier et al., 2021). The effect of recent warming and drought on area burned is also exacerbated due to the fuels accumulated in many areas as a result of century-long fire suppression efforts (Marlon et al., 2012; Parks et al., 2015).

Incidences of large and severe fires often result in severe environmental and social impacts, such as: poor air quality (O'Dell et al., 2019; Xie et al., 2022), negative health effects from smoke exposure (Burke et al., 2022), enhanced streamflow (Williams et al., 2022), increased flood and debris risk (Jong-Levinger et al., 2022), major vegetation shifts in ecosystems (Coop et al., 2020), and mass displacement of human populations (Jia et al., 2020). Moreover, to manage these fires, federal firefighting expenditures in the United States soared from $\sim \$0.5$ billion in the late 1980s to an average of $\sim \$3$ billion between 2016 and

2021 (source: https://www.nifc.gov/fire-information/statistics/suppression-costs). Thus, understanding the complex, multiscale interactions between climate, vegetation, and human drivers of wildfire activity is of vital scientific and social importance.

Individual wildfire events in the WUS are caused by the coincidence of fire conducive hot and arid weather in presence of adequate vegetation and sources of ignition (Parisien and Moritz, 2009; Williams and Abatzoglou, 2016). However, the influence of specific climatic conditions such as high temperatures and low precipitation may vary spatially due to the fuel

moisture content, biomass distribution, and local topography in flammability-limited regions such as forests (Westerling, 2016), and temporally through the response of vegetation growth to antecedent conditions in fuel-limited regions such as grasslands and shrublands (Swetnam and Betancourt, 1998). Meanwhile, the larger WUS fires typically burn over a period of several weeks or more, so the climatic effect on total area burned is regulated by short-term fire weather conditions such as prolonged temperature and aridity extremes (Gutierrez et al., 2021; Juang et al., 2022), sustained intense wind events over multiple

days (Potter and McEvoy, 2021), or even the continuity provided by fuels within a landscape's heterogeneous vegetation structure (Rollins et al., 2002). Although difficult to model precisely, fire regimes across the WUS are also affected by the spatial variability of lightning strikes (Romps et al., 2014; Kalashnikov et al., 2022) and stochastic human ignition patterns (Balch et al., 2017; Keeley and Syphard, 2018; Keeley et al., 2021). When aggregated over multiple wildfire events, the observed trends in fire frequency and total area burned carry imprints of the nonlinear, spatially heterogeneous, temporally

integrated interactions between climate, vegetation, human, and topographic variables. Physical models of wildfire activity in the WUS, consequently, require a wide suite of input predictors over multiple spatiotemporal scales to accurately represent the various dynamical processes that promote or inhibit fire ignitions and growth.

Here we focus on statistical models for two important fire variables, frequency and area burned. Broadly, these models infer the empirical relationships between observed wildfire activity at a given spatiotemporal scale and its various climate,

vegetation, and human drivers. To account for the multiple degrees of freedom characteristic to the problem, regression based models tend to study the mean state relationship between wildfire activity and its drivers by averaging all variables along spatial (Abatzoglou and Williams, 2016) or temporal dimensions (Parisien and Moritz, 2009; Parisien et al., 2012). Despite





being instrumental in clarifying the role of different fire drivers on large spatiotemporal scales, these, and similar, analyses are unable to model fire activity at smaller scales that are important for allocating fire suppression and rescue resources or
identifying regions for preventive fuel treatment. On the other hand, other efforts based on classical (Westerling et al., 2011) and Bayesian (Joseph et al., 2019) statistical methods as well as machine learning (ML) approaches (Coffield et al., 2019; Jain et al., 2020; Wang and Wang, 2020; Wang et al., 2021; Joshi and Sukumar, 2021; Kuhn-Régnier et al., 2021; Kondylatos et al., 2022) have modeled grid-scale fire activity across various spatial extents. Besides representation of finer-scale processes, another key advantage of the grid-scale analyses over the mean state approach is their ability to determine the hierarchy of
important wildfire drivers at various spatiotemporal scales.

In this paper, we introduce a stochastic ML (SML) model, SMLFire1.0, to estimate the probability distributions of monthly fire frequencies and sizes in $12\,\mathrm{km} \times 12\,\mathrm{km}$ grid cells across the WUS based on data from 1984 to 2020. SMLFire1.0 consists of a pair of mixture density networks (MDNs) constructed by appending a custom loss function (Ebert-Uphoff et al., 2021) to a neural network. We adopt the MDNs to determine the parametric distributions of fire frequencies and sizes using a combination
of static and dynamic climate, vegetation, human, and topographic predictors. We then simulate fire frequencies for each grid cell as well as sizes for grid cells with non-zero frequencies. Our results are visualized and discussed at broader spatial scales of ecoregions for ease of comparison with results from previous analyses.

Our modeling approach for SMLFire1.0 builds upon and extends previous methods in four important ways: *a)* unlike other ML methods based on gradient boosted trees or quantile regression, our use of parametric distributions in SMLFire1.0, es-
pecially for individual fire sizes, provides a straightforward way to implement uncertainty quantification for our predictions; *b)* we account for the spatiotemporal variability of the predictors and their nonlinear interactions; *c)* our model includes fire frequencies and locations while simulating the total area burned, thus enabling projections of total area burned for different idealized future scenarios of fuel flammability, human ignition patterns, and fuel treatment; *d)* the combined frequency and size ML framework serves as a single model across the entire WUS and does not require separate training for predicting fire
activity in each constituent region. While we do not explore the scenario in detail here, the flexibility and efficiency of our ML framework also makes it an ideal subgrid-scale parameterization scheme for the fire modules of regional scale Dynamic Vegetation Models (DGVMs) (Li et al., 2012; Rabin et al., 2017) as well as Earth System Models (ESMs) (Zou et al., 2020).

## 2   Data

### 2.1   Study Ecoregions and Divisions

Our study region consists of all $12\,\mathrm{km} \times 12\,\mathrm{km}$ grid cells in the continental US west of $103°$ W longitude. We visualized the results of our analysis at the Bailey's Level III (L3) ecoregion (Bailey, 1996) scale for clarity and ease of comparison, especially in terms of interannual variability, with prior results in the literature (Abatzoglou et al., 2017; Williams et al., 2019; Joseph et al., 2019). Moreover, in several analyses (Littell et al., 2009; Parisien and Moritz, 2009; Dennison et al., 2014), organizing the study region in terms of ecoregions or ecoprovinces has been useful in identifying the broad contours of climate-fire
relationships. We define an "Ecoregion" to be constituted by one or more similar L3 ecoregions to ensure sufficient statistics



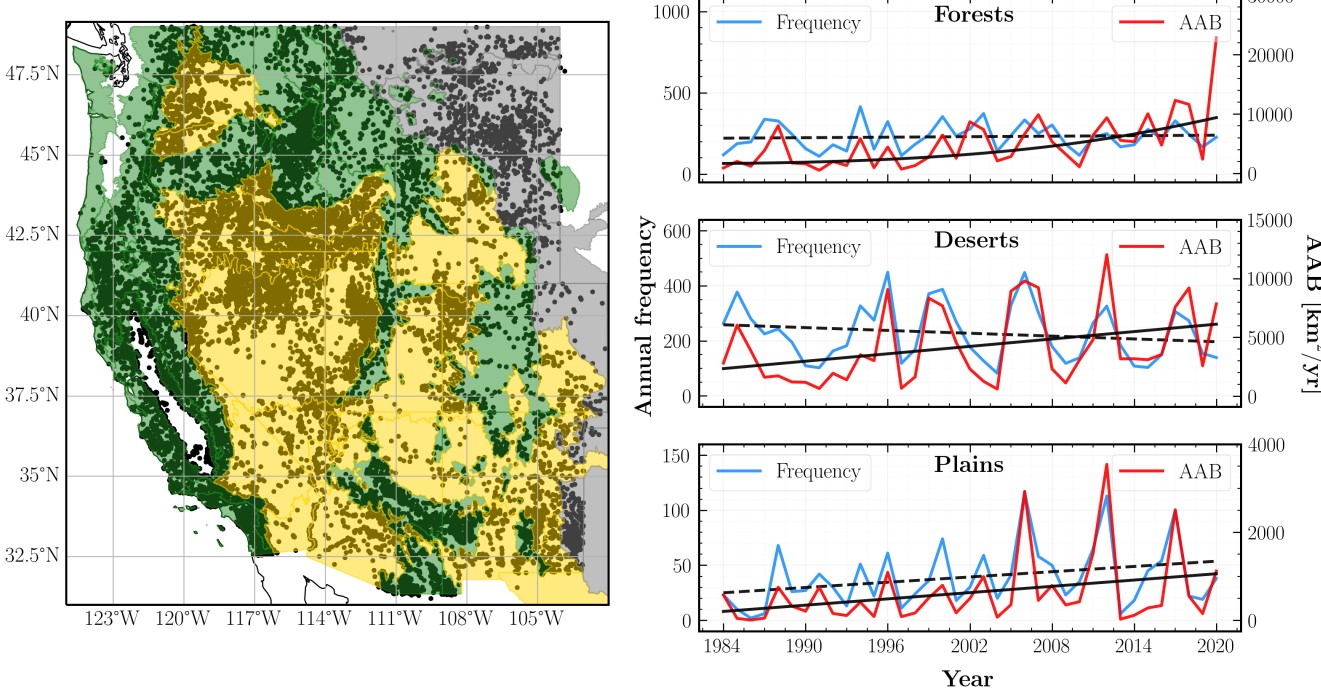

**Figure 1.** Wildfire activity in the western United States (WUS) from 1984 to 2020. *Left:* locations of fire centroids (black dots) across the WUS with the spatial extent of three ecological Divisions characterized by their primary vegetation type – Forests (green), Deserts (yellow), and Plains (gray). *Right:* Observed annual fire frequencies (blue) and annual area burned (AAB) (red) for each Division. The black curves indicate the statistically significant trends for each AAB (solid) and annual frequency (dashed) time series.

(refer to Table S1 for more details), considering a total of 18 Ecoregions across the western United States for this study. Further, we follow (Brey et al., 2018) and organize our Ecoregions in terms of three broad ecological "Divisions" that are characterized by their primary vegetation types, namely Forests, Deserts, and Plains. Note that all three Division types consist of a combination of both forested and non-forested areas albeit in different proportions.

## 2.2 Wildfire Activity

We focus on two primary fire variables in this analysis: occurrences and sizes. Both these variables are available in the Western US MTBS-Interagency (WUMI) wildfire dataset (Juang et al., 2022) that contains 18646 fire locations and burned areas from 1984 to 2020. The recently released WUMI dataset (accessed Sep 12, 2022) is a collection of unique fires $\geq 4\,\mathrm{km}^2$ from the Monitoring Trends in Burn Severity (MTBS) program (Eidenshink et al., 2007) and fires $\geq 1\,\mathrm{km}^2$ from the following federal agencies: California Department of Forestry and Fire Protection (CalFire), US Fish and Wildlife Service (FWS), US Forest Service (FS), Bureau of Indian Affairs (BIA), Bureau of Land Management (BLM), Bureau of Reclamation (BOR), and the National Park Service (NPS). Notably, the WUMI dataset underrepresents fires $\leq 4\,\mathrm{km}^2$ from 2018–2020, especially



in non-forested areas outside of California because of missing post-2017 data from the BLM, BIA, BOR, and NPS (source: https://famit.nwcg.gov/applications/FireAndWeatherData/ZipFiles; accessed Sep 24, 2022). Although fires smaller than $4\,\mathrm{km}^2$

have a negligible contribution to the total area burned, they constitute $\sim 50\%$ of all fires in our study domain. Thus, the artificially low frequency of smaller fires in 2018-2020 as represented in the current version of the WUMI database likely hinders with accuracy with which our current modeling effort can simulate the probability of small fires.

In Fig. 1, we map all WUMI fire locations as well as plot time series of annual frequency and annual area burned (AAB) time series for the Forests, Deserts, and Plains Divisions along with their statistically significant trends. The AAB trends are

evaluated using a least squares linear regression fit to the log transformed area burned time series as in Williams et al. (2019).

## 2.3 Input Predictors

We consider four broad classes of input predictors – three dynamic plus one static – aggregated to the $12\mathrm{km}\times12\mathrm{km}$ grid scale: climate and fire weather, vegetation, human-related (henceforth human), and topographic. At this spatial scale, a vast majority of fires ($\sim 97\%$) have sizes smaller than the size of the grid cell. Choosing a finer resolution would require explicitly modeling

the spatial autocorrelation between the burned area in neighboring grid cells, whereas a coarser resolution results in lower accuracy while correlating fire properties to its environmental variables.

We select six primary climate and fire weather predictors: temperature, precipitation, vapor pressure deficit (VPD), snow water equivalent (SWE), wind speed, and lightning. Monthly climate grids for mean daily maximum temperature (Tmax), daily minimum temperature (Tmin), and precipitation total (Prec) are taken from National Oceanic and Atmospheric Admin-

istration's (NOAA) Climgrid dataset (Vose et al., 2014); additionally, gridded dew point temperatures for computing VPD are adapted from PRISM (Daly et al., 2004). We consider two additional fire danger predictors which have been shown to significantly correlate with fire activity (Abatzoglou and Kolden, 2013): 1,000-hr fuel moisture (FM1000), and the Fosberg Fire Weather Index (FFWI) (Fosberg, 1978); these were both derived from specific combinations of the primary predictors with daily meteorological grids from gridMET (Abatzoglou, 2013). Furthermore, we use daily scale data from the UCLA

ERA5-WRF reanalysis (Rahimi et al., 2022) to calculate the monthly maximum $X$-day mean of daily maximum and minimum temperature ($\mathrm{T_{maxX}}$, $\mathrm{T_{minX}}$), where $X \in \{3, 7\}$. Similar $X$-day extreme predictors are also derived for VPD, FFWI, and wind speed. The monthly mean and maximum daily SWE variables come from the gridded National Snow and Ice Data Center (NSIDC) dataset (Zeng et al., 2018, 2019). Antecedent conditions often exhibit significant correlations with fire activity through drying of soils and fuels as well as promoting fuel growth over multiple months (Westerling et al., 2006; Wang and

Wang, 2020; Abolafia-Rosenzweig et al., 2022). Thus, for a given fire month $m$, we include temperature, precipitation, VPD, and SWE based predictors that are running averages of monthly mean values from month $m-1$ to $m-t$, where $t \in \{2, 3, 4\}$. We also include the mean annual precipitation for each of the two years prior to the fire year ($\mathrm{AntPrec_{lag1}}$ and $\mathrm{AntPrec_{lag2}}$) as additional predictors to probe long drought legacy effects (Bastos et al., 2020; Wu et al., 2022). An important source of ignitions over a large area of the WUS is lightning, most frequently as part of summer thunderstorms. We use the Vaisala

National Lighning Detection Network (NLDN) lightning strike density data (Wacker and Orville, 1999; Orville and Huffines,





2001) aggregated to monthly scale with coverage from 1987 to 2020. For all months between December 1983 and January 1987, we assume monthly climatological means for the missing lightning data.

We leverage land type data from the National Land Cover Dataset (NLCD) (Yang et al., 2018) for deriving annual scale vegetation predictors. Since the NLCD classifies land cover type (e.g., evergreen forest, cropland etc.) across the US at a 30m
spatial resolution, we calculate the fraction of each 12x12km grid cell occupied by a given NLCD land-cover classification. The NLCD is not an annual product and provides maps of landcover classification for: 1992, 2001, 2004, 2006, 2008, 2011, 2013, 2016, and 2019. For years between two NLCD years, the landcover in each grid cell is linearly interpolated between the NLCD years, whereas for years before or after 1992-2019, landcover is assumed to be the same as the nearest NLCD year. We adopt three predictors: Grassland, Shrubland, and Forest, each of which represents the fraction of landcover in a grid cell
covered by the respective vegetation type. Besides the fraction of landcover, we include a more direct representation of fuel abundance through the aboveground biomass map from (Spawn et al., 2020). Although the biomass map (Biomass) is available for only one year, 2010, we justify its inclusion by positing that the spatial variability of vegetation across the WUS is more dominant than the temporal variability of the vegetation in a majority of grid cells. Thus, for all modeling purposes, we treat Biomass as a static predictor. In future work, it will be ideal to include simulated vegetation biomass maps (e.g., (Hansen et al.,
2022)) in a coupled framework within the wildfire model.

Combining the following NLCD land cover types that reflect the presence of urban areas: "Developed High", "Developed Low", "Developed Medium", and "Developed Open" we construct a single, annual scale human predictor, Urban. For more granular information of human settlements, we include predictors for human population (Popdensity), defined in terms of distance from the nearest area with population density greater than 10 people per square kilometer, and mean housing
density (Housedensity). These predictors are adapted to annual timescales using data for three years: 1990, 2000, and 2010 from the SILVIS dataset (Radeloff et al., 2005) following the same interpolation procedure that we used for NLCD predictors. Other static human predictors include: mean number of camp grounds (Camp_num), mean distance to nearest camp grounds (Camp_dist), and the distance to nearest highway (Road_dist) derived from publicly available datasets (source: https://www.naturalearthdata.com/downloads/10m-cultural-vectors/roads/; http://www.uscampgrounds.info/takeit.html). Lastly, to
incorporate the effect of topography on fire activity (Holsinger et al., 2016; Harris and Taylor, 2017), we include two static variables: mean slope (Slope) and mean south-facing degree of slope (Southness). Altogether we include a total of 51 potential predictors, and summarize their names, identifiers, temporal scale, and sources in Table S2.

Before analyzing the data with a statistical model, we perform an additional preprocessing step. To account for spatiotemporal heterogenity of the WUS ecological landscape, we "standardize", *i.e* subtract the mean and divide by the standard deviation,
all input predictors. Dynamic predictors, including all climate and most vegetation variables, at each location are standardized in time, whereas the static predictors are standardized across the entire spatial domain.

## 3  Model Description





## 3.1 Theory

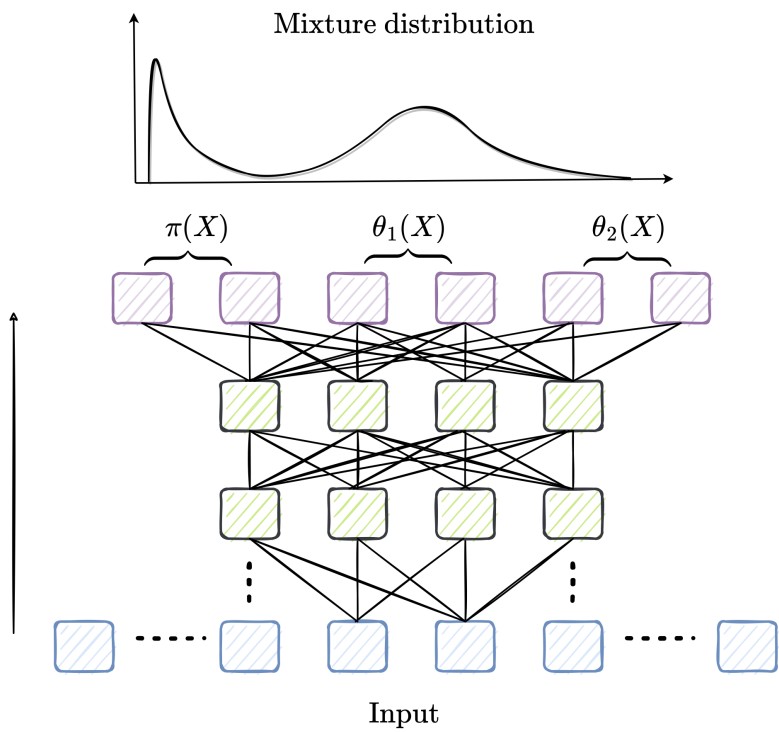

**Figure 2.** Schematic diagram illustrating the input (blue), hidden (green), and output (purple) layers of a Mixture Density Network (MDN) model within the SMLFire1.0 framework. While a fully connected neural network is implemented in practice, only a partial connected one is shown here for clarity; the solid black line on the left denotes the direction from the input to the output layer, whereas the dotted black lines represent additional nodes and layers in the network. Also shown above the output layer are the parameters for a two component mixture distribution of the form given in Eq. 1.

Our main goal is to develop a statistical model for fire frequency and sizes as a function of input predictors described in
the previous section. Specifically, we want our model to: *a)* capture the nonlinear, spatially heterogeneous interactions among the climate, vegetation, human, and topographic variables that influence wildfire activity; *b)* rely on physical variables and be independent of location and time of year; *c)* be based on parametric distributions that may be sampled to generate uncertainty estimates of the modeled values. We build upon previous efforts (Westerling et al., 2011; Joseph et al., 2019; Wang et al., 2021) by constructing a model that combines the flexibility of machine learning techniques with the robustness of parametric
distributions.

In SMLFire1.0, we use two mixture density networks (MDNs) to separately model the conditional probability (henceforth conditional for brevity) distributions for fire frequency and sizes on a monthly time scale. A MDN is a fully connected, feedforward neural network whose output layer consists of parameters of a mixture model (Bishop, 1994). In other words, we



use a neural network with multiple hidden layers, illustrated in Fig. 2, to map the nonlinear functional relationship between
different predictor variables and output data onto the parameters of a mixture of standard statistical distributions. A mixture
distribution is a useful tool for representing the probability distribution of outputs with multiple modes or peaks. Thus, given
observed data $Y$, we learn the functional mapping between input predictors $X$ and output parameters $\psi$ by minimizing a loss
function of the general form,

$$\mathcal{L}(Y|X; \psi \in \{\pi, \theta\}) = \prod_{n=1}^{N} \left( \sum_{m=1}^{M} \pi_m(X_n) p_m(Y_n|\theta_m(X_n)) \right); \quad \sum_{m=1}^{M} \pi_m = 1, \tag{1}$$

where $M$ and $N$ denote the number of mixture components and data points respectively, and each mixture component consists
of a conditional distribution $p_m(\theta_m)$ as well as a weight parameter $\pi_m$. To ensure that the resultant mixture distribution is
normalized, we constrain the sum of all individual weight parameters to be 1.

We use the monthly fire counts (including zeros) in each grid cell across the WUS as the data for our fire frequency model.
In total, we consider data in about $\sim 10$ million grid cells out of which only 17489 correspond to observed fires. Common
choices of parametric distributions for representing count data include: binomial, Poisson, and negative binomial distributions.
All of these distributions are often also used in conjunction with another distribution, as part of a zero-inflated mixture model,
that accounts for the additional zeros in the data coming from an independent process such as fire suppression. In this analysis,
for each space-time grid cell (henceforth grid cell), $n$, we use a zero-inflated Poisson lognormal distribution (ZIPD) to model
the observed fire frequencies $f_n$ as a function of the input predictors $X_n$,

$$\mathcal{L}(f_n|X_n; \pi, \mu, \delta) = \begin{cases} \pi(X_n) + (1 - \pi(X_n)) \operatorname{Pois}(f_n|\mu(X_n), \delta(X_n)), & , f_n = 0; \\ (1 - \pi(X_n)) \operatorname{Pois}(f_n|\mu(X_n), \delta(X_n)) & , f_n > 0, \end{cases} \tag{2}$$

where $\pi$ is the probability of an independent process that generates zeros, and the rate parameter of the Poisson (Pois) distribu-
tion is drawn from a lognormal distribution with mean $\mu$ and variance $\delta^2$. There are two major challenges to this approach for
modeling fire frequencies: *a)* a large proportion of grid cells contain no fires, resulting in a significant data imbalance problem,
*b)* minimizing the risk of missing fires in our predictions, we tend to overpredict fires in grid cells that saw no fires, leading to
a high false positive rate. To address point *a)*, we experiment with both downsampling, *i.e.* considering only a random subset
of all available grid cells with no fires similar in size to the number of observed fires, and upsampling, *i.e.* generating multiple
duplicate samples of the observed fires to match the size of grid cells with no fires, to address this imbalance in our analy-
sis. On the other hand, to fix the effects of a high false positive rate due to the large number of non-fire grid cells, we use a
spline regression model for calibrating the mean and variance of the predicted frequencies to those of the observed data at the
Ecoregional scale. Finally, we aggregate the predicted fire frequencies across all grid cells within an Ecoregion and compute
the mean and variance of the fire frequency, $F$, for a given month, $l$, as follows,

$$\mathbb{E}[F^l|X] \equiv \sum_{n \in \mathrm{L3}} \mathbb{E}_{\mathcal{L}_n(\hat{\pi}, \hat{\mu}, \hat{\delta})}[f_n^l|X_n^l],$$

$$\mathrm{Var}[F^l|X] \equiv \sum_{n \in \mathrm{L3}} \mathrm{Var}_{\mathcal{L}_n(\hat{\pi}, \hat{\mu}, \hat{\delta})}[f_n^l|X_n^l], \tag{3}$$





where $\mathbb{E}_{\mathcal{L}_n(\cdot)}[\cdot]$ and $\mathrm{Var}_{\mathcal{L}_n(\cdot)}[\cdot]$ indicate the expected value, or mean, and variance with respect to the conditional frequency distribution given by Eq. 2, and the hats denote the distribution parameters fixed to their optimal values determined by training

the MDN. The expected value and variance are evaluated using Monte Carlo (MC) simulations of the frequency distribution at monthly and annual timescales.

Meanwhile, as shown previously (Schoenberg et al., 2003; Littell et al., 2009; Li and Banerjee, 2021), wildfire sizes across several spatial scales follow a probability distribution with large tails, or equivalently an extreme value distribution, such that a majority of fires are small but the total area burned is dominated by a small number of large fires. In this analysis, we consider

three extreme value mixture distributions for fire sizes: Generalized Pareto distribution (GPD), Lognormal distribution, and a composite Lognormal-Generalized Pareto distribution (Lognormal-GPD) (Scollnik, 2007). The Lognormal-GPD is included to account for the possibility that the fire sizes follow a hybrid distribution with a pronounced hump as well as a significant tail, which are the features of the Lognormal distribution and GPD respectively. We consider 9953 fires from the WUMI dataset with sizes greater than a threshold of $4\,\mathrm{km}^2$ for the GPD and Lognormal-GPD fire size models, whereas we consider all fires

for the Lognormal case as it does not require a threshold for extreme events. Due to our fixed choice of the grid for input predictors, a majority of the fires have burned areas that are spread across two or more grid cells. This raises the question: which values of the input variables should we consider as predictors for our model? Approximating each fire as a circle with area equal to its size, we consider the "effective" input predictors for a given fire to be the average of inputs over all grid cells intersected by the fire's perimeter weighted by the fraction of burned area in each grid cell. We model each fire $j$ as independent

draws from a conditional mixture distribution (Carreau and Bengio, 2007) of the general form,

$$\mathcal{L}(A_j|X_j;\pi,\theta_1,\theta_2) = w(A_j) \sum_{m=1}^{M} \pi_m(X_j) p_m(A_j|\theta_{m,1}(X_n),\theta_{m,2}(X_j)), \tag{4}$$

where $(\theta_1,\theta_2)$ are the parameters of a heavy tailed distribution determined by the MDN for each fire. For the GPD, $(\theta_1,\theta_2)$ represent the scale and concentration parameters, whereas for the Lognormal case they represent the mean and standard deviation of the distribution's natural logarithm. We include a weight factor $w(A_j)$, that is inversely proportional to the frequency

of size $A_j$ in the training data, to account for data imbalance due to the relative disparity in the number of small and large fires.

The conditional distributions of monthly and annual area burned (MAB and AAB respectively) are obtained by aggregating the distribution of fire sizes for each grid cell in an Ecoregion with a fire. We compute the mean and variance for the fire size distributions using MC simulations and formulas similar to the ones described in Eq. 3. We interpret the expressions for MAB and AAB as follows: assuming that the mean size of all fires at a given spatiotemporal scale $l$ are identical and denoted by

$A$, the mean total area burned $A_l$ is simply given by $\mathbb{E}[A_l|X_l] = \mathbb{E}[F_l|X_l] \times A$. Phrased differently, the expected area burned at a given spatiotemporal scale is linearly proportional to the mean fire frequency, $\mathbb{E}[F_l|X_l]$ with a constant coefficient $A$. We note, in practice, that since the mean fire size of the GPD model is similar for most fires, the mean fire frequency also plays an important role in determining the mean MAB or AAB.

Following Iglesias et al. (2022), we assume that $A$, the mean size of an individual fire, as well as the total size distribution,

could be time-dependent, or nonstationary, in general. We allow for an enhancement or weakening in the response of fire sizes to one or more predictors by including a nonstationary response in the size model. In particular, we use the entire training





dataset to construct two models: one model with a reweighted GPD loss function (GPD-Ext) for the time period with larger fire sizes, and the other one for the remaining years with an unweighted loss function (GPD). We stitch these models together with a breakpoint to construct the combined GPD MDN model.

In order to isolate the role of frequency in the total area burned, we first derive the AAB using the combined GPD MDN model evaluated with observed fire frequencies and values of input predictors corresponding to the observed locations of fires given in the WUMI dataset. We also explore three further variations for the WUS AAB time series: one, using modeled frequencies for each Ecoregion from the frequency MDN model with observed fire locations; second, with observed frequencies but input predictors corresponding to model fire locations predicted by the frequency MDN; and third, with both fire frequencies

and locations drawn from the frequency MDN model. Since our modeled frequencies also include smaller fires, we apply an additional time-dependent scaling factor to account for the relative abundance of large fires ($\geq 4\,\mathrm{km}^2$) while deriving the area burned with modeled frequencies.

    Lastly, to obtain percentiles of the burned area distribution we require the full probability density function defined over all grid cells with fires,

$$p(A^l|X, N^l) = p_{A_1^l|X_1^l} * \cdots * p_{A_{N^l}^l|X_{N^l}^l} \;; \quad p_{A_j^l|X_j^l} = \mathcal{L}(A_j^l|X_j^l, f_j^l > 0; \hat{\pi}_j^l, \hat{\theta}_{1;j}^l, \hat{\theta}_{2;j}^l). \tag{5}$$

where $*$ denotes the convolution operator, and $N^l$ are the number of fires at a given spatiotemporal scale. Rather than solving this expression analytically (Nadarajah et al., 2018), we sample it with MC simulations and report the $0.5^{\text{th}}$, $50^{\text{th}}$ and $99.5^{\text{th}}$ percentiles of the monthly and annual area burned at the WUS and Divisional scales.

## 3.2    Implementation

We implement our SMLFire1.0 framework using the `Keras` interface for `TensorFlow` library version 2.7.0. During training, we allow our neural networks to have the following tunable hyperparameters: number of hidden layers, $n_l$; number of neurons per layer, $n_{ne}$; and the number of mixture components, $n_c$. We use the relevant distributions available in the `TensorFlow Probability` library version 0.15.0 for designing custom loss functions for both the frequency and size data.

    For model training, we hold out 1 contiguous year (which we take to be 2020, unless specified otherwise) of fires and input predictors as test data, and split the monthly data from the remaining 36 years $70\%$ to $30\%$, chosen randomly, between training and validation data. The out-of-sample validation data is useful for evaluating model performance, which we measure through several metrics defined in the section below.

    We train our model for up to 500 epochs using the Adam optimizer with learning rate, `lr_rate` $= 10^{-4}$, to learn the optimal distribution parameters for each model. Since a typical MDN with several hidden layers consists of $\sim \mathcal{O}(1000)$ hidden

weights and biases, it is plausible that the model overfits the training data; we address this issue by applying three regularization steps: early stopping of the training process when the model performance does not improve for 10 epochs; L2 regularization, which constrains the squared sum of all the network weights; and a Dropout layer which randomly sets a fraction, `dr_frac`, of inputs to 0 to improve co-learning of the remaining weights. Based on numerical experiments in the pre-training phase,





we fix the L2 regularization rate to be `reg_rate` $= 10^{-3}$, and the dropout fraction to be `dr_frac` $= 0.4$. Altogether, the regularization procedure reduces overfitting and improves generalization for the MDN.

### 3.3 Metrics

We define metrics for two broad purposes: enabling model selection and measuring model performance. For the former, a straightforward choice is the value of the loss function, given in Eq. 1, averaged across all batches and epochs: reducing the loss improves the model. As our model size is small, we ignore the effects of model complexity that may be included through metrics such as the Akaike Information Criterion (AIC). Since we are predicting a mixture distribution rather than a point estimate of the output, we use a variation of the Kullback-Leibler (KL) divergence to measure the distance between distributions,

$$D'_{\mathrm{KL}}(C_\psi || \hat{C}) \equiv \sum_{n=1}^{N} \log \left( \frac{C_\psi(y_n | x_n)}{\hat{C}(y_n | x_n)} \right); \quad \text{Accuracy} = 100 \times (1 - D'_{\mathrm{KL}}(C_\psi || \hat{C})), \tag{6}$$

where $C_\psi$ is the cumulative distribution function (CDF) of fire frequencies or sizes predicted by the respective MDN, and $\hat{C}$ is the empirical CDF estimated from data. Unlike the ratios of the probability density functions used in calculating the classic KL divergence, we adopt a ratio of the CDFs which is an equivalent estimator in the asymptotic limit (Perez-Cruz, 2008). We also perform leave-one-out cross validation (LOO-CV) to approximate the model's generalization error. Specifically, we create subsets of fires by excluding the fires that occurred in 1 contiguous year for all years in our study period and report the validation accuracy averaged over all such subsets as the LOO-CV score.

Having selected a model, we characterize its performance by measuring, statistically, how well or poorly the modeled time series fits the observed data. We use the Pearson's correlation coefficient, $r$, to gauge the proportion of variance in the observed data explained by the predicted time series. Moreover, to account for the point-wise uncertainty that we obtain from our MC simulations, we use the chi-squared statistic,

$$\chi^2 = \sum_{n \in \mathcal{S}_T} \frac{(y_n - \hat{y}_n)^2}{\hat{\sigma}_n^2}, \tag{7}$$

as a measure of the goodness-of-fit for all frequencies or area burned at a particular spatiotemporal scale, $\mathcal{S}_T$. To ensure uniformity of scale, we report the reduced chi-squared statistic, $\chi_r^2$, which is the chi-squared value defined above divided by the degrees of freedom. In our case, the degrees of freedom are simply given by the number of years with non-zero values minus the number of parameters.

### 3.4 Predictor importance

We estimate the sensitivity of model output to input predictors using the SHapley Additive exPlanation (SHAP) technique (Lundberg and Lee, 2017). SHAP values are a recent approach (see Wang et al. (2021) for an application to fire modeling) to characterize the marginal contribution of each input predictor on individual predictions by using a game-theoretic approach to account



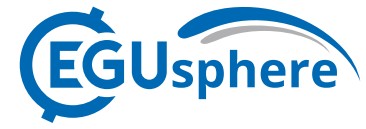

| Fire variable | MDN type | $(n_l, n_{ne}, n_c)$ | Loss | Accuracy | LOO-CV score |
|---|---|---|---|---|---|
| Frequency | ZIPD (Upsampled) | (2, 16, 2) | 3.29 | 87.53 % | 75.75 % |
| | ZIPD (Downsampled) | (2, 16, 2) | **0.85** | **95.67**% | 94.18 % |
| Size | GPD | (3, 8, 2) | **4.22** | **91.71**% | 90.60 % |
| | Lognormal distribution | (3, 8, 2) | 4.45 | 91.39 % | 86.92 % |
| | Lognormal-GPD | (2, 16, 4) | 4.36 | 87.15 % | 85.49 % |

**Table 1.** Summary of the mixture density network (MDN) architecture and performance metrics used for modeling fire frequencies and sizes in SMLFire1.0. $n_l$, $n_{ne}$, and $n_c$ refer to the number of hidden layers, neurons per layer, and the number of mixture components respectively. The loss is dimensionless for the frequency MDN whereas the loss for size MDN has units of inverse area burned, or $\mathrm{km}^{-2}$. Both the loss and accuracy metrics are reported for validation data, whereas the leave-one-out cross validation (LOO-CV) score is the validation accuracy averaged over different subsets of fire years, where each subset is obtained by excluding all fires in a randomly chosen year. Lower values of loss as well as higher values of accuracy and LOO-CV score indicate better model performance.

for the contributions of all possible coalitions of the remaining predictors. This is in contrast to the traditional predictor importance techniques which only rely on a fixed coalition of predictors to assess the contribution of an individual variable, and are therefore susceptible to correlations between input predictors. We implement the SHAP technique in our analysis by adapting the `KernelExplainer` method from the `shap` python package (source: https://github.com/slundberg/shap).

A SHAP value $s$ for an input predictor $p$ can be interpreted as follows: $p$ contributes $s$ additional units to the model output determined by combining the mean baseline value along with the contributions of all other predictors. To estimate the sensitivity of our model outputs to various predictors in a particular Ecoregion, we first randomly choose a subset of grid cells within that Ecoregion with no observed frequencies, or background points, to compute our mean baseline SHAP value. Then combining a fraction of the background points with grid cells that have observed fires in a fixed ratio to create a pool of test points, we evaluate the SHAP values for all predictors relative to the mean baseline value at each test point. The choice of the ratio does not affect our results as long as the number of background points constitute a minority fraction of the test points. For the results shown in the following section, we use a 1:3 ratio of background to test points for each Ecoregion to ensure sufficient statistics. In total, we evaluate the SHAP values for all input predictors at $\sim 20,000$ test points across the WUS.

We visualize our results using two types of plots: a summary plot that shows the SHAP values of the leading input predictors at each test point colored by the predictor value alongside the partial dependence plots of two important predictors, and a global feature importance plot of the leading predictors ordered according to their mean absolute SHAP values, or $S$, for each Ecoregion. We also assess the interaction effect between predictors by applying a color gradient of one predictor's values to all test points in the partial dependence plots of the other. However, since all test points do not explicitly include information about vegetation transitions under climatic perturbations in the respective grid cell or the effect of repeated fire burns, the indicated predictor importance and partial dependence plots are valid only under the assumption of a stationary relationship between the input predictors and fire sizes.





## 4 Results

### 4.1 Model Selection

The first step in our model selection process is determining the optimal hyperparameter configuration for each loss function. We train the frequency and size MDNs on a subset ($\sim 40\%$) of the overall training data over a grid of hyperparamater configurations. In particular, for both the frequency MDNs as well as the Lognormal-GPD size MDN, we fix the number of components, $n_c = 2$, while varying the number of hidden layers, $n_l$, the number of neurons per layer, $n_{\mathrm{ne}}$; for the rest of the cases, we vary $n_c$ as well. The model performance is evaluated using the three metrics defined above: average loss, maximum accuracy, and the LOO-CV score computed over all the epochs. The optimal configuration for a MDN type is defined as the one with the lowest loss and the highest validation accuracy. Since our choice of validation data as a random subset of the training data (as opposed to selecting data for consecutive years) already serves as a form of cross-validation, the LOO-CV score may be interpreted as a measure of the model's mean performance across different initial conditions as it is computed with different subsets of validation data. The second step is reducing the number of predictor variables by iteratively dropping all predictors that do not improve overall model performance and are highly correlated ($r \geq 0.5$) with other predictors. We include all fire month and static predictors in this step, using the iterative procedure to identify the most important antecedent and extreme weather permutations of each relevant climate variable. After this step we are able to narrow our predictor basis from 51 to 28 variables.

The optimal hyperparameters and performance metrics for each MDN are outlined in Table 1. For the fire frequency model, we indicate results for the ZIPD MDN trained with upsampled and downsampled data separately. We find that the downsampled ZIPD MDN performs slightly better than its upsampled counterpart, despite the latter using more data (see Chatterji et al. (2022) for a discussion of an analogous problem in the ML literature). Among the size models, we find that the GPD MDN has the best performance and prefers only two components, while the optimal Lognormal distribution MDN configuration contains four components albeit with a higher loss and lower validation accuracy. The optimal Lognormal-GPD MDN has an intermediate performance relative to those of the GPD and Lognormal distributions. We also calculated the LOO-CV score with 3 contiguous years of data held out and find that there is only a marginal decline in model performance, highlighting the robustness of our ML models. In the following sections, unless stated otherwise, we refer to the downsampled ZIPD and GPD MDNs as the frequency MDN and size MDN respectively.

### 4.2 Fire Frequency

We use a MDN trained on downsampled training data to determine the parameters of the ZIPD for fire frequencies in each grid cell across the WUS from 1984 to 2020. MC simulations of the parametric frequency distributions for all grid cells are aggregated to compute the mean fire frequency and its $2\sigma$ uncertainty intervals over monthly and annual time scales. These are plotted for the entire study domain in Fig. 3 as well as individual Ecoregions selected on the basis of total fire counts and their quality of fit in Figs. 4 and 5. The frequencies are plotted for both monthly and annual timescales and are contrasted with the observed values at the respective spatial scale. We note that the observed fire frequency between 2018 and 2020 could





increase after including several missing smaller fires in the WUMI dataset, which may also potentially affect our modeled frequencies. We evaluated the goodness-of-fit between our predictions and observations at the annual timescale through the Pearson's correlation coefficient and the reduced chi-squared statistic shown in Table 2 for all the WUS Ecoregions.

An upshot of our likelihood-based MDN model in SMLFire1.0 is the availability of uncertainty estimates (Riley and Thompson, 2016) for the predicted fire frequencies. Since our frequencies aremodeled as a Poisson distribution, we expect their standard deviation to scale as $\sim \sqrt{N}$ for $N$ fire counts. Thus, the relatively narrow 95% error band shown for both the WUS and regional frequency plots comes with an important caveat: we only estimate the statistical uncertainty for our results while ignoring the (possibly dominant) contribution to the model uncertainty from sources such as climate-vegetation linkages (Kitzberger et al., 2017; Zhou et al., 2019; Bastos et al., 2020; Tschumi et al., 2022).


At the WUS level, our mean modeled frequencies are in good agreement with the total number of observed fires, exhibiting high correlations at both monthly ($r = 0.94$) and annual ($r = 0.85$) timescales. Our model also successfully captures the interannual variability and monthly extremes across most of the Ecoregions. In particular, the modeled annual frequencies for

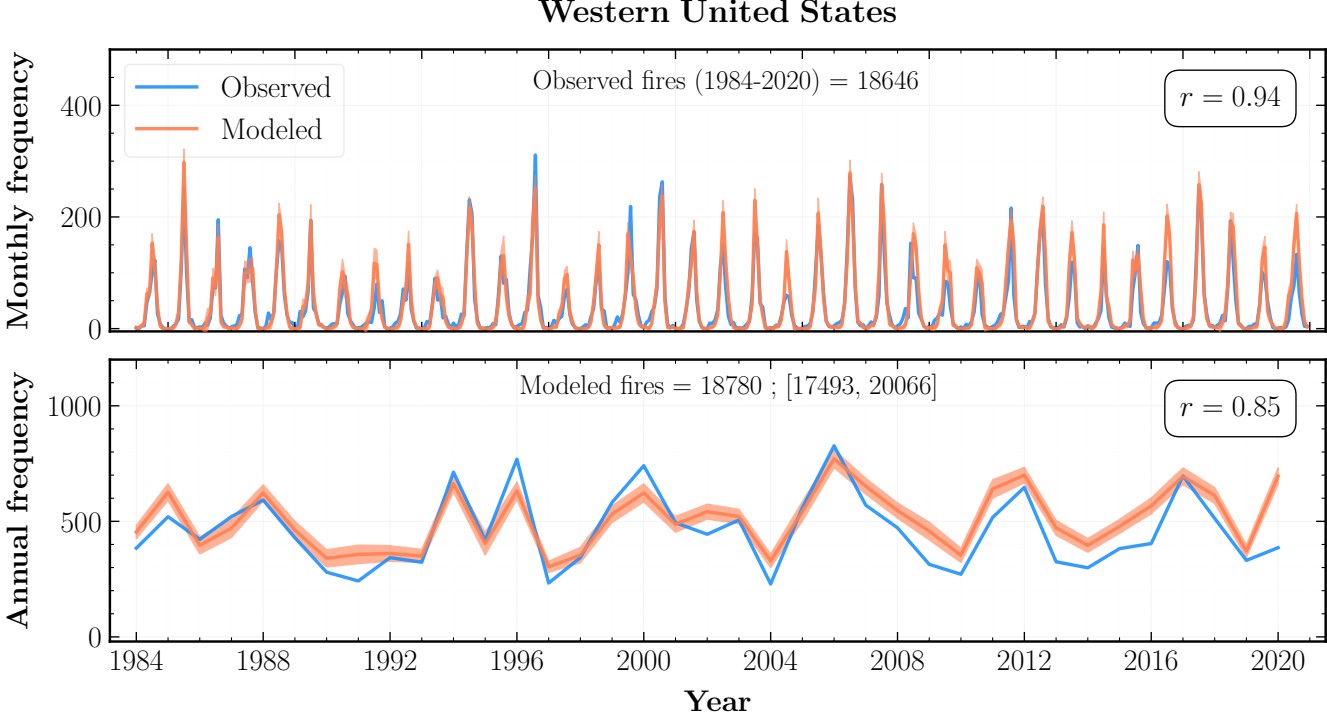

**Figure 3.** Observed (blue) and modeled (orange) fire frequencies across the western United States at monthly and annual time scales from 1984 to 2020. Orange shaded regions indicate $2\sigma$ uncertainty intervals for the mean fire frequency aggregated over the Monte Carlo (MC) simulations for all grid cells. The mean number of modeled fires over the study period as well as its $2\sigma$ uncertainty interval are indicated at the top of the lower panel. Also shown within each subplot is the Pearson correlation coefficient ($r$) between the observed and modeled time series.



**Figure 4.** Observed (blue) and modeled (orange) fire frequencies at monthly and annual scales from 1984 to 2020 for Ecoregions selected based on the total number of fires and goodness-of-fit metrics. The orange shaded regions within each subplot indicate $2\sigma$ uncertainty intervals for the mean regional fire frequency aggregated over the Monte Carlo (MC) simulations for all constituent grid cells. Also shown is the Pearson correlation coefficient ($r$) between the observed and modeled fire frequency time series for each Ecoregion.





**Figure 5.** As in Fig. 4, but with different WUS Ecoregions.




| Division | Ecoregion | Frequency | | Size | |
|---|---|---|---|---|---|
| | | $r$ | $\chi^2_r$ | $r$ | $\chi^2_r$ |
| Forests | Sierra Nevada | 0.70 | 20.98 | 0.60 | 0.84 |
| | California (CA) North Coast | 0.68 | 107.75 | 0.58 | 1.52 |
| | CA Central Coast | 0.46 | 75.64 | 0.60 | 1.17 |
| | CA South Coast | 0.61 | 40.91 | 0.66 | 1.70 |
| | Pacific Northwest Mountains | 0.70 | 9.63 | 0.81 | 0.66 |
| | Northern Rockies | 0.89 | 11.28 | 0.93 | 0.34 |
| | Middle Rockies | 0.85 | 25.69 | 0.84 | 1.30 |
| | Southern Rockies | 0.82 | 10.20 | 0.79 | 0.81 |
| | Arizona/New Mexico Mountains | 0.72 | 45.57 | 0.63 | 1.18 |
| Deserts | American (AM) Semidesert | 0.88 | 15.80 | 0.95 | 0.77 |
| | Intermountain (IM) Semidesert | 0.66 | 36.52 | 0.90 | 0.43 |
| | IM Desert | 0.80 | 34.99 | 0.92 | 0.39 |
| | Chihuahuan (CH) Desert | 0.62 | 68.93 | 0.91 | 0.52 |
| | Columbia Plateau | 0.78 | 14.79 | 0.77 | 0.56 |
| | Colorado Plateau | 0.71 | 12.10 | 0.72 | 0.39 |
| | Southwestern (SW) Tablelands | 0.67 | 15.88 | 0.94 | 0.71 |
| Plains | Northern Great Plains | 0.87 | 5.75 | 0.93 | 0.43 |
| | High Plains | 0.65 | 8.73 | 0.94 | 0.57 |

**Table 2.** Goodness-of-fit metrics in terms of Pearson's correlation ($r$) and the reduced chi-squared statistic ($\chi^2_r$) between the observed and modeled time series for both frequencies and area burned at an annual time scale. Both $r$ and $\chi^2_r$ are dimensionless metrics; higher values of $r$ and lower values of $\chi^2_r$ indicate a better fit. Results are shown for each Ecoregion organized by their ecological Division.

Sierra Nevada, Pacific Northwest Mountains, Northern, Middle, and Southern Rockies among the Forests Division; American
Semidesert, Intermountain (IM) Desert, Columbia and Colorado Plateaus among the Deserts Division; and the Northern Great Plains are strong correlated ($r \geq 0.7$) with the observed frequencies.

Broadly, the trends in fire frequencies can be characterized as a competition between three independent drivers. One, an increasing trend in climate dryness (Seager et al., 2015; Abatzoglou and Williams, 2016), which is correlated with regional water balance and hence fuel flammability; two, better communication of fire risk resulting in fewer accidental ignitions (Keeley and
Syphard, 2018) and enhanced preparedness levels; and three, increased human fire suppression efforts through improvements in fire prevention and containment techniques. While the WUMI dataset indicates moderate increases in the annual number of wildfires $> 1\,\mathrm{km}^2$ in areas defined as forest by NLCD, this is almost fully compensated by a reduction in frequency of fires in non-forested areas. As a result, there is no clear trend in the observed fire frequency for the WUS. On the other hand, our modeled frequencies indicate a mildly increasing trend at the overall WUS scale as well as for several Ecoregions such as PNW



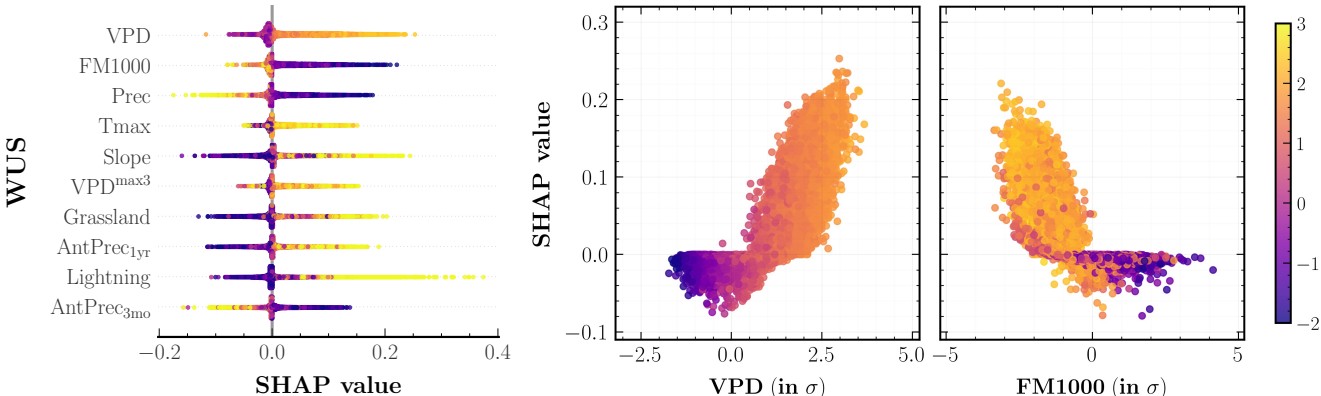

**Figure 6.** SHapley Additive exPlanation (SHAP) analysis of the fire frequency MDN model outputs across the western United States. *Left:* Input predictors sorted in descending order of their mean SHAP values aggregated over the entire study period. Each colored point along the $x$-axis represents an individual prediction with the color corresponding to high (yellow) or low (indigo) values of the respective input predictor. *Middle* and *Right:* Partial dependence plots for two important predictors shown on the $x$-axis, colored corresponding to high (yellow) or low (indigo) values of the mean daily maximum temperature, Tmax. The colorbar (far right) is normalized in terms of standard deviations ($\sigma$s) for all relevant values across the three panels.

Mountains, Columbia Plateau, and IM Desert. A potential contributing factor to this variability could be the high sensitivity of fire frequencies to hot and dry conditions in our model combined with the inadequate representation of human action. The latter is important especially since human predictors such as population and housing density can have a dual effect on fire frequencies: proximity to urban settlements increases the probability of ignitions and access to suppression resources, whereas reductions in fuel continuity due to development and land management drastically reduce the probability that an individual

ignition grows into a large wildfire (Knorr et al., 2014; Andela et al., 2017).

Since our model is trained on data over the whole WUS, its performance, on average, is better over Ecoregions with larger number of fires, such as the Middle Rockies and IM Semidesert. On the other hand, our model performs quite poorly for regions with a low number of total fires, where it is more likely to exhibit large interannual variations over a baseline of very few to no fires. This behavior is evident in the plot for CH Desert in Fig. 5 as well as the low $r$ and high $\chi_r^2$ values in Table 2

for Ecoregions such as CA North Coast, SW Tablelands, CH Desert, and High Plains.

The SHAP values for individual predictors of the frequency MDN as well as the partial dependence plots for two important predictors, VPD ($S = 0.042$) and FM1000 ($S = 0.030$), with a color gradient corresponding to Tmax values are plotted for the WUS in Fig. 6. Here, $S$ denotes the mean absolute SHAP values of each predictor. Similar plots at the Divisional level for Forests, Deserts, and Plains are shown in Fig. 7. We also include the mean SHAP plots for each of the 18 Ecoregions considered

in our analysis in Figs. S2 and S3. The main drivers of fire frequencies at all spatial scales are climate predictors correlated with hot and arid fire weather conditions. Thus, high VPD is the leading predictor for most Ecoregions, most frequently combined with low fuel moisture in large diameter fuels, FM1000. Other important predictors include Tmax, Prec, and Slope, such that



**Figure 7.** SHapley Additive exPlanation (SHAP) analysis of the fire frequency MDN model outputs for different western United States' Divisions: (top) Forests, (middle) Deserts, and (bottom) Plains. *Left column:* Input predictors sorted in descending order of their mean SHAP values aggregated over the entire study period. Each colored point along the $x$-axis represents an individual prediction with the color corresponding to high (yellow) or low (indigo) values of the respective input predictor. *Middle*, *Right:* Partial dependence plots for two important predictors shown on the $x$-axis, colored corresponding to high (yellow) or low (indigo) values of the mean daily maximum temperature, Tmax (top and middle panel), and high (indigo) or low (yellow) values of the 1000-hr dead fuel moisture, FM1000 (bottom panel). The colorbar (far right) is normalized in terms of standard deviations ($\sigma$s) for all relevant values across the three panels.





high daily maximum temperatures, lower fire month precipitation total, and higher mean slope all contribute to higher fire frequency on average.

Among the other subdominant predictors, antecedent climate conditions play a varying role across different Divisions. Antecedent snowpack estimated using 3 month average SWE, $\mathrm{AvgSWE_{3mo}}$, is an important predictor especially in the Rocky Mountains Ecoregions ($S = 0.023$), with lower SWE in 3-4 antecedent months leading to higher predicted frequencies in the fire month. Increases in modeled fire frequencies for several Ecoregions in Deserts and Plains are also driven by antecedent conditions at both the seasonal and annual timescales through lower values of $\mathrm{AntPrec_{3mo}}$ and higher values of $\mathrm{AntPrec_{1yr}}$

predictors. The latter result corroborates previous analyses (Crimmins et al., 2004; Abatzoglou et al., 2017) which have highlighted the role of high prior year precipitation in promoting fuel growth within arid regions where vegetation is often too limiting to allow for large fires. The spatial variability in vegetation predictors, however, is of low importance for most Ecoregions. Interestingly, for similar dryness levels, our model simulates more fires for sites with lower values of Biomass relative to sites with higher Biomass in Forests; meanwhile, higher fraction of grassland results in a higher fire frequency across all three

Divisions. Our model considers lightning strike density as an important predictor across several Ecoregions, most notably over the CA North ($S = 0.021$) and South ($S = 0.024$) Coasts, PNW Mountains ($S = 0.021$), and Middle Rockies ($S = 0.019$). Human predictors, on the other hand, are not among the top 10 predictors for any Ecoregion and therefore of little overall relevance to the frequency model.

We visualize the response of fire frequencies to individual predictors through the partial dependence plots in Figs. 6 and 7.

SHAP values for all four variables shown in the plots: VPD, FM1000, Prec, and $\mathrm{AntPrec_{3mo}}$ exhibit near-linear relationships above a threshold with their respective predictor values. The color gradient of all test points in Forests and Deserts shows that VPD and FM1000 are strongly correlated with Tmax, while also highlighting the interaction effect between Tmax and Prec. In Plains, instead of Tmax, we consider the interaction effect of FM1000 on VPD and $\mathrm{AntPrec_{3mo}}$ to explore the influence of antecedent and fire month predictors on fuel moisture. FM1000 values exhibit a strong interaction effect with antecedent

precipitation in our model, but not with fire month VPD since fuel moisture shows significant correlations with atmospheric aridity. In other words, SHAP values for our frequency model vary with Prec and $\mathrm{AntPrec_{3mo}}$ predictors only for sites with high values of Tmax and low values of FM1000.

### 4.3   Fire Size

We use MDNs trained on fires $\geq 4 \ \mathrm{km}^2$ to determine the parameters of the combined GPD and GPD-Ext (henceforth combined

GPD) distribution of individual fire sizes. MC simulations of the parametric size distributions for all observed fires from 1984 to 2020 are aggregated to compute the mean of the monthly and annual area burned (MAB and AAB respectively) and their $1\sigma$ uncertainty intervals. The total MAB and AAB simulated using the combined GPD model with a breakpoint after 2004 are plotted for the entire WUS in Fig. 8 and separately at the Ecoregion level in Figs. 9 and 10. The goodness-of-fit metrics, namely Pearson's correlation and reduced chi-squared statistic, between the predicted and observed sizes for each

Ecoregion are summarized in Table 2. We plot the SHAP values for individual predictors at test points across the WUS in Fig. 12, alongside partial dependence plots for two important fire size predictors, VPD and Grassland, with a color gradient



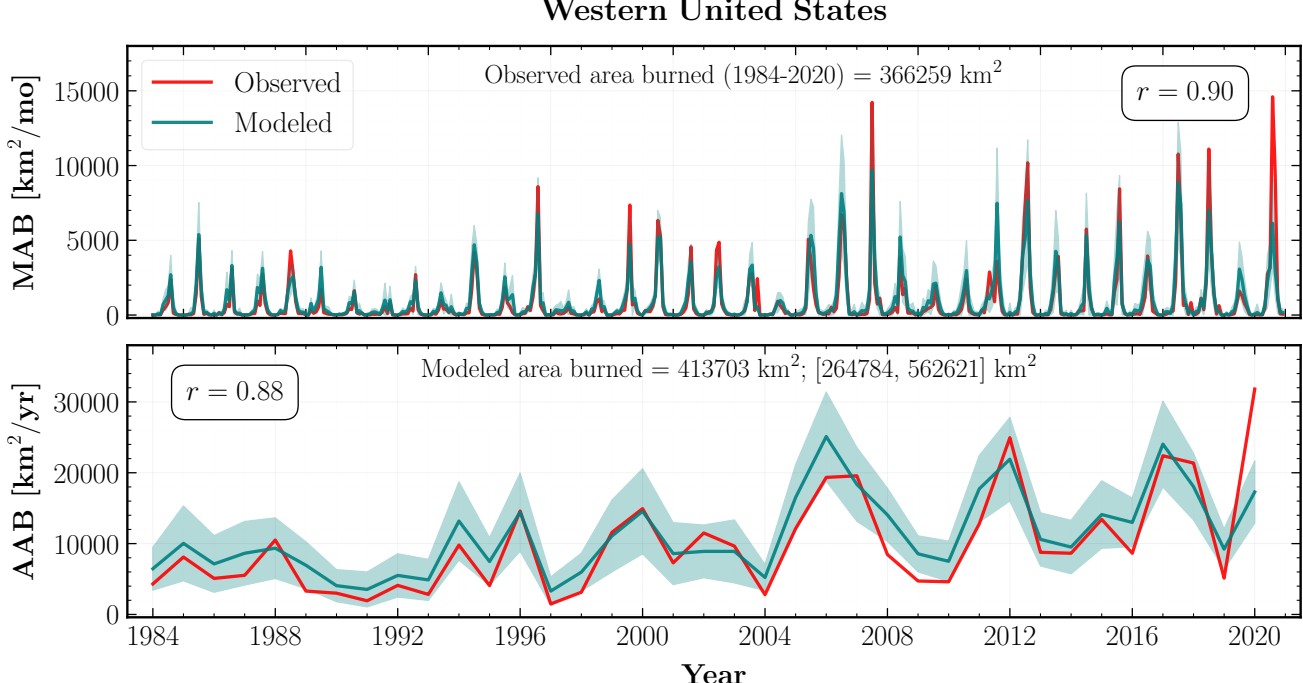

**Figure 8.** Observed (red) and mean modeled (teal) area burned across the western United States at monthly (MAB) and annual (AAB) time scales from 1984 to 2020. The teal shaded regions indicate $1\sigma$ uncertainty intervals for the mean area burned aggregated over the Monte Carlo (MC) simulations from the combined GPD model for all observed fires. The mean total area burned over the study period as well as its $1\sigma$ uncertainty interval are indicated at the top of the lower panel. Also shown within each subplot is the Pearson correlation coefficient between the observed and modeled burned area time series.

corresponding to Tmax values. These plots are constructed using a procedure similar to the one described in the previous section for fire frequencies. We also show the partial dependence plots at the Division level are in Fig. 13 and plot the mean SHAP values for individual Ecoregions in Figs. S7 and S8.

The introduction of a time-dependent response and a breakpoint after 2004 in our modeling is justified through the following analysis. As indicated by Fig. 8, there is a rising trend in the AAB for the WUS with significantly more large MAB months $\sim 2000$ onward than in 1984-1999. Moreover, as shown by Juang et al. (2022), this increase in AAB is driven by the exponential response of fire size to atmospheric aridity and not due to increasing fire frequency. We confirm their result by noting that the complementary cumulative distribution function (CCDF) of observed fire sizes between 1984-2004 is markedly different from
the CCDF for sizes observed between 2005-2020 as plotted in Figs. S4 and S5. In fact, as shown in Fig. S4 (right panel), the CCDF of the combined GPD model is in much better agreement with the observed CCDF than the CCDFs of either model individually (Fig. S4; left and middle panels). After varying the breakpoint for different years between 2000 and 2006, we find





**Figure 9.** Observed (red) and mean modeled (teal) monthly (MAB) and annual area burned (AAB) from 1984 to 2020 for the Ecoregions shown in Fig. 4. The teal shaded regions within each subplot indicate $1\sigma$ uncertainty intervals for the mean regional area burned aggregated over the Monte Carlo (MC) simulations from the combined GPD model for all observed fires. Also shown is the Pearson correlation coefficient ($r$) between the observed and modeled area burned time series for each Ecoregion.



**Figure 10.** As in Fig. 9, but with different WUS Ecoregions.





that a breakpoint after 2004 results in the best fit to the observed area burned. Thus, we successfully model the shift to larger fire sizes observed after 2004 by including an additional GPD distribution with fatter tails.

We emphasize that the improved agreement between the CCDFs of observed and modeled sizes is not merely an artifact of the breakpoint procedure; in fact the choice of the distribution plays a critical role. Specifically, we verify this by repeating our analysis with the Lognormal distribution, which has thinner tails than the GPD. In Fig. S5, we demonstrate that the CCDF of the reweighted Lognormal MDN (Lognorm-Ext) underestimates large portions of the observed sizes' CCDF while being able to account for only the most extreme fires. Consequently, the combined distribution (Lognorm-Comb) is an inadequate model

for the observed fire sizes over the study period.

The choice of fire frequencies – either observed or modeled – and the stochasticity in fire locations affects both the inter-annual variability and total area burned of the modeled AAB time series. Contrasting the results shown in Fig. 8 and Fig. S6, we note that the AAB using modeled frequencies and observed locations results in a moderate decrease in the total area burned along with a marginal improvement in the correlation with the observed AAB. Further, simulating fire sizes with model

locations leads to a significant rise in the total area burned irrespective of the frequency source, although the AAB time series with observed frequencies has a notably weaker correlation ($r = 0.77$) compared to the case with modeled frequencies ($r = 0.91$). We explain this behavior with respect to both fire locations and frequencies. First, the frequency MDN tends to locate fires in grid cells with extreme values of input predictors such as VPD and Prec, leading to simulated fire sizes larger than those at observed fire locations; important fire predictors at the latter, especially before 2004, having high (or low), but not

extreme, values. Second, as shown in Fig. 3, the modeled frequencies are consistently lower than the observed ones between 1993 and 2001 leading to a lower simulated area burned relative to the case with observed frequencies for the same source of fire locations. At the same time, the anomalously high modeled frequency for 2020 leads to improved correlations for the modeled AAB time series irrespective of the choice of fire locations. As a cross-check, we compute the correlation coefficient for the two AAB time series simulated with observed locations and the observed AAB between 1984 and 2019, and find that

the simulation with observed frequencies has a higher correlation ($r = 0.93$) than the one with modeled frequencies ($r = 0.89$). We do not find a similar improvement in the correlations of the two AAB time series simulated with model locations. These results serve as an important lesson for modeling fire activity: *improved correlations of model predictions with observed data may not always be a good indicator of improved model accuracy*.

Using the combined GPD MDN model, our modeled MAB ($r = 0.90$) and AAB ($r = 0.88$) time series are very good fits

to the observed area burned, within the $1\sigma$ uncertainty interval, when considering the entire WUS. Our model performs well across most of the WUS with AAB predictions for 15 out of the 18 Ecoregions exhibiting strong correlations ($r \geq 0.7$) with the observed area burned. Moreover, as shown in Figs. 9 and 10, the trends in the modeled AAB time series successfully emulate the distinct multidecade increases in observed AAB over both forested and non-forested Ecoregions.

Our model has mixed skill in predicting large MAB and AAB during the study period. For example, our model is able to

simulate the full range of AAB variability in the Northern Rockies, Northern Great Plains (Fig. 9; top and middle left panels), and American Semidesert (Fig. 10; middle right panel), but it fails to capture the largest AAB between 1984 and 2019 in Middle Rockies, IM Semidesert (Fig. 9; top and middle right panels), and PNW Mountains (Fig. 10; top right panel). This




tendency holds even after reweighting the size distribution for post-2004 fires. In particular, the total AAB during the year
2020 deserves further scrutiny. The modeled AAB significantly underestimates the observed 2020 AAB in Fig. 8, predicting

only about half of the observed value. We see similar behavior at the Ecoregional level in Columbia Plateau (Fig. 9; middle left panel), Sierra Nevada, and Southern Rockies (Fig. 10; top and middle left panels).

With these results, we can make a stronger assessment about our modeling framework: first, for almost all years in our study period, the mean of the aggregate area burned distribution is a good approximation for the observed time series, so the only challenging part is determining the time dependence of the mean sizes of individual fires; and second, while the discrepancy

between modeled and observed area burned in 2020 highlights a clear limitation of our model, can we use still use it to make meaningful predictions for anomalous extreme fire years?

In Fig. 11, we show the total modeled AAB for two extreme fire years with the largest area burned, 2012 and 2020, at both the WUS and Divisional scales. We find that the observed AAB for 2012 is in the $\sim 80^{\text{th}}$ percentile of the predicted WUS AAB, driven primarily by the large AAB in Deserts. However, for 2020, the observed $\gtrsim 99.5^{\text{th}}$ percentile Forests AAB

resulted in the total WUS AAB to be in the $99.5^{\text{th}}$ percentile of our model predictions. Contrasted with previous extreme fire years, 2012 and 2017 (see Fig. S9), the observed 2020 AAB is: *a)* in very high percentiles of the modeled AAB simulated with observed frequencies, implying that the observed fire sizes for 2020 were much greater than those in the 1984-2019 period that

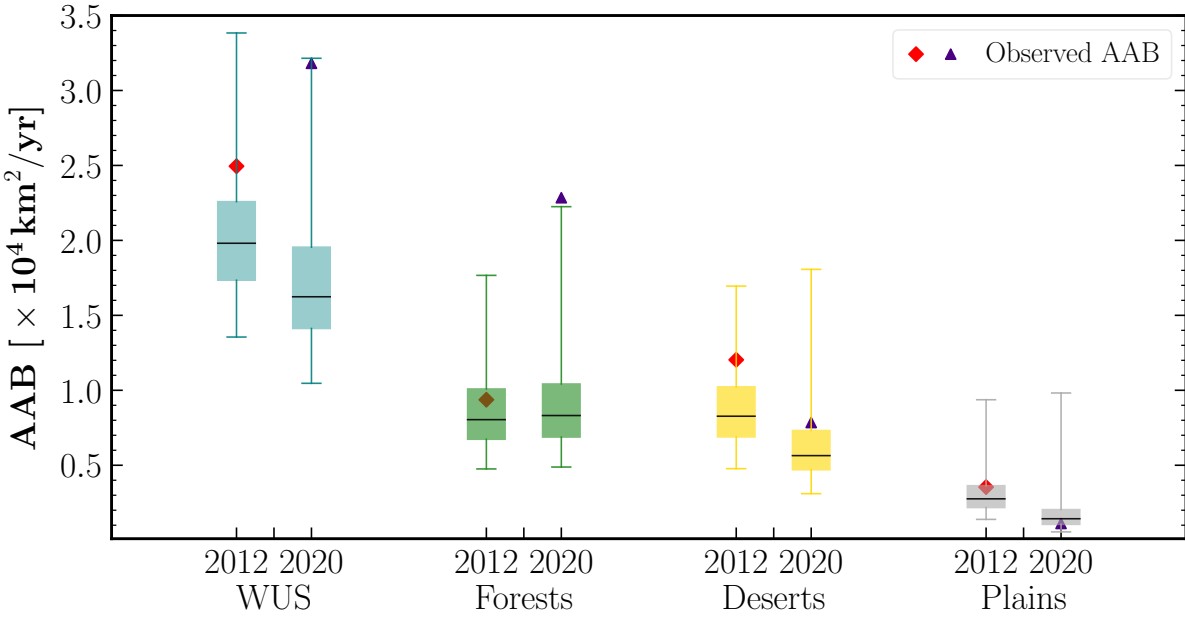

**Figure 11.** Boxplots of predicted annual area burned (AAB) for two extreme fire years, 2012 and 2020, for the entire western United States (WUS) (teal) and three Divisions organized by their primary vegetation types: Forests (green), Deserts (yellow), and Plains (gray). The lower and upper whiskers of each boxplot indicate the $0.5^{\text{th}}$ and $99.5^{\text{th}}$ percentile of the predicted AAB distribution, whereas the black line represents its median value. Also shown for reference are the observed AAB for both 2012 (red diamond) and 2020 (indigo traingle).





our model is trained on; *b)* driven by anomalously high AAB in the Forests Division, and is a striking example of an extreme fire year driven by large fires in flammability-limited areas rather than fuel-limited ones.

Among the 10 input predictors of fire size shown in descending order of importance in Fig. 12, the SHAP technique selects VPD ($S = 3.95$), Grassland ($S = 2.60$), and FM1000 ($S = 2.34$) as the three most important predictors at the WUS level. Again, $S$ refers to the mean absolute SHAP value for each predictor. These are also among the top predictors at the Divisional scale as shown in Fig. 13, with Grassland being more important than FM1000 in Deserts ($S = 2.75$) and Plains ($S = 3.33$). Broadly, SHAP values for all predictors besides FM1000, Prec, and $\mathrm{AntPrec_{3mo}}$, have a positive relationship with higher
predictor values. Another important predictor at the WUS and Forests level is Slope ($S = 2.43$): its SHAP values indicate that fire size is promoted by large topographic slope, which is consistent with previous findings (Andrews, 2018). Assessing the predictor importance at the Ecoregion level, as illustrated in Fig. S7 and S8, we find that climate and fire weather predictors are dominant drivers across Forests, whereas Grassland plays a larger role in Deserts and Plains. The importance of grassland cover could also signal the role of invasive grass species (Knapp, 1998; Balch et al., 2013) in driving large area burned within
our model. Thus, vegetation plays a much more important role in simulating area burned for the size model as compared to the frequency model. This is also true for most Ecoregions in Forests Division, where the spatial distribution of aboveground biomass serves as an important secondary predictor. The mean absolute SHAP values suggest that weekly scale extreme weather predictors such as $\mathrm{FFWI^{max3}}$ are also important predictors in several Ecoregions. We interpret the response of fire sizes simulated by our model to the climate at different temporal scales as follows: monthly to seasonal scale hot and arid
weather create favorable conditions for fire spread, while the growth of large fires is facilitated by weekly scale extreme fire weather (Jacobson et al., 2022). The SHAP plot for Popdensity in Deserts ($S = 1.54$) and Plains ($S = 1.32$) indicates that higher predictor values result in simulation of larger fire sizes. This could be because increased distance from populated areas is correlated with a decrease in accessibility for fire suppression efforts, and therefore higher occurrences of larger fires in the observed record.

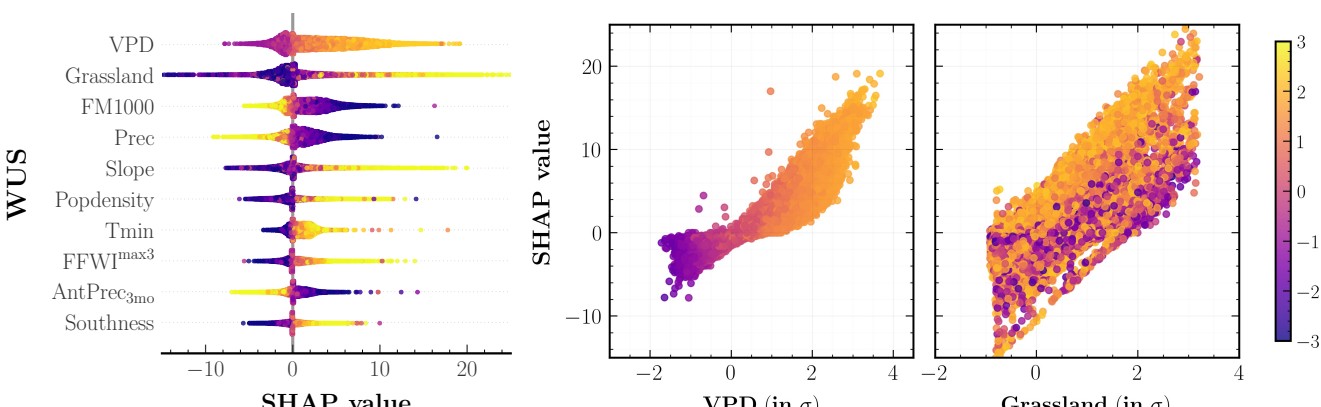

**Figure 12.** As in Fig. 6, but for the fire size MDN model.



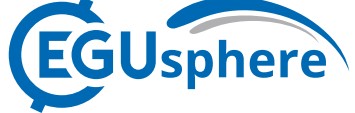

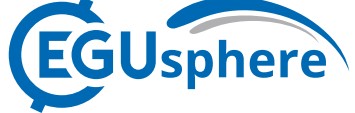

**Figure 13.** SHapley Additive exPlanation (SHAP) analysis of the fire size MDN model outputs for different western United States' Divisions: (top) Forests, (middle) Deserts, and (bottom) Plains. *Left column:* Input predictors sorted in descending order of their mean SHAP values aggregated over the entire study period. Each colored point along the $x$-axis represents an individual prediction with the color corresponding to high (yellow) or low (indigo) values of the respective input predictor. *Middle, Right:* Partial dependence plots for two important predictors shown on the $x$-axis, colored corresponding to high (yellow) or low (indigo) values of the mean daily maximum temperature, Tmax (all panels). The colorbar (far right) is normalized in terms of standard deviations ($\sigma$s) for all relevant values across the three panels.





We also retrain our fire size model with two different variations of the input predictors selected for the main analysis: first with relative humidity (RH), average RH over 3 antecedent months ($\mathrm{AvgRH_{3mo}}$), and 3 day minimum RH ($\mathrm{RH^{min3}}$) instead of VPD and its corresponding permutations; and second including all VPD and RH predictors. There are no significant differences in the correlations between the simulated and observed WUS AAB time series in either case, but the SHAP summary plots shown in Fig. S10 provide valuable insights. In the first case where VPD is not considered as a potential predictor, RH

($S = 1.67$) replaces VPD as the leading fire size driver across the WUS, and $\mathrm{AvgRH_{3mo}}$ ($S = 1.00$) is more important relative to other antecedent climate predictor. When allowing both VPD and RH to serve as predictors, VPD ($S = 3.58$) has higher predictive power than RH ($S = 1.95$) in our model at both the WUS and Forests Divisional scales. The fact that the decision of including VPD, RH, or both does not substantially affect model performance does not mean that this decision is unimportant. As Brey et al. (2021) point out, VPD is projected to continue rising dramatically while projected RH decreases are more mod-

erate. In this paper we prioritize the model that uses VPD, because VPD is more directly representative of the atmosphere's evaporative demand (Anderson, 1936; Monteith, 1965).

    Lastly, we show the responses of fire sizes to individual predictor values for all test points at the WUS and Divisional level in Figs. 12 and 13 respectively. We find that fire sizes simulated by our model respond, above a threshold, exponentially to increases in VPD and decreases in FM1000 at all spatial scales, although the response is notably stronger in Forests. This

result is consistent with the findings of Juang et al. (2022) who showed that the exponential response of fire sizes to increasing aridity appears to arise from the fact that large fires have much greater capacity for area growth than smaller fires. Meanwhile, we do not find any significant interactions between Tmax and VPD as well as FM1000; Grassland shows a weak interaction effect with Tmax, such that sites with the same fraction of grassland cover yield larger sizes at higher mean daily maximum temperatures.

## 5   Discussion

We have developed a novel stochastic ML framework, SMLFire1.0, for modeling fire activity across different WUS Ecoregions. Although the fire frequency and area burned time series simulated using this framework are in good agreement with observations at multiple spatial and temporal scales, there are several areas of improvement across three interconnected themes: modeling approach and architecture, vegetation, and other potential predictors. We discuss each one of these themes in detail

below.

    – *Modeling:* A limitation of the frequency model is that we are, effectively, estimating a joint distribution between ignitions and fire likelihood. In other words, we are using data for observed fires, which occur randomly, to learn the relationships between different predictors that contribute to fire conducive conditions. However, such an approach may introduce a bias in ignition-limited regions that could have large fire-prone areas with no fire occurrences (Parisien and

545         Moritz, 2009). One way to improve our framework would be to model ignitions using spatial stochastic processes, or to compute fire probabilities using a presence-only approach (Chen et al., 2021). We also expect that further improvements to the WUMI dataset, especially for smaller fires, would improve the accuracy of our modeled frequency time series.





For the fire size model, we combined two GPD distributions with a breakpoint after 2004 to obtain a distribution that best fit the cumulative distribution of observed fire sizes. Rather than introduce a breakpoint by hand, in future work we intend to explore and model the mechanisms that may have led to such a distribution shift. At the computational level, we plan on incorporating a recurrent ML component in the current neural network architecture in order to model the nonstationary fire size responses over multiple timescales. An extension would be to expand the parametric size distribution by including a smooth, differentiable form of the Lognormal-GPD with an arbitrary threshold parameter. Using such a hybrid distribution would ensure that our model has more flexibility in simulating a large number of small fires from a distribution with finite mean and variance as well as a small number of larger fires from a distribution with finite mean but infinite variance (Cohen and Xu, 2015). To improve model interpretability and avoid post hoc evaluations of feature importance such as SHAP (Hooker et al., 2021), a more robust alternative would be to build an interpretable ML model from the bottom up as outlined in (Alvarez-Melis and Jaakkola, 2018).

– *Vegetation:* A major area of desired improvement for SMLFire1.0 is the representation as well as the dynamic structure of vegetation predictors. This could be done in several different ways. First, by including finer scale vegetation characteristics through a combination of integrated data products, such as Effective Vegetation Type (EVT) (Rollins, 2009) or Normalized Difference Vegetation Index (NDVI) (Didan, 2015), and outputs from physically parameterized models (Hansen et al., 2022). These predictors would be helpful in informing the model about the type and spatial distribution of different live and dead fuels. Second, for predictions of future fire activity over longer time scales, it would be important to account for the nonstationarity of the climate-vegetation relationship, a pivotal factor in determining the spatially heterogeneous shifts in vegetation patterns (Higuera et al., 2009; Bradstock, 2010; Hansen et al., 2018). We may already be seeing evidence of this effect in our analysis: recent increases in aridity coupled with transitions in vegetation patterns could have precipitated a shift in the fire regimes across the WUS and promoted larger and more severe fires in the past two decades. Third, besides climate induced shifts, vegetation patterns are also affected by human and natural disturbances such as changes in land use(Klein Goldewijk and Ramankutty, 2004), tree mortality (Williams et al., 2013), insect range expansions as well as infestations (Pureswaran et al., 2018), and fire itself (Parks et al., 2018). Importantly, multiple studies have shown that fire-induced fuel limitations are expected to slow, but not abate the continued heat- and drought-induced increases in annual area burned across the WUS over the next few decades (Hurteau et al., 2019; Abatzoglou et al., 2021a). Thus, coupling the current stochastic ML model framework with a DGVM for a variety of vegetation types and different human intervention scenarios is a promising research direction.

– *Other predictors:* Several potentially relevant land-surface predictors were not considered here since their records are not available over the full duration of our study period. For instance, recent work has highlighted the role of remotely sensed soil moisture (Rigden et al., 2020) and the sensitivity of live fuel moisture content to atmospheric aridity (Rao et al., 2022) in regulating wildfire ignitions and area burned respectively. Reliable measurements over the WUS for both these predictors are only available after 2015. Meanwhile, human influence on individual fire sizes could be affected





by synchronous fire activity over several regions. Abatzoglou et al. (2021b) approximate this effect by concurrent fire danger days, a metric that measures the strain on available resources for suppressing new ignitions as well as containment of ongoing fires. We intend to explore the role of additional land-surface and human action predictors in forthcoming analyses.

## 585  6  Conclusions

Disentangling the various climate, vegetation, and human drivers of wildfire frequency and sizes in the western United States is critical for developing accurate seasonal as well as longer term forecasts of fire activity. In this paper, we introduced a novel stochastic ML framework, SMLFire1.0, for estimating the parametric distributions of observed fire frequencies and sizes in $12\mathrm{km} \times 12\mathrm{km}$ grid cells observed on a monthly time scale. The parametric distributions were sampled using Monte Carlo
simulations to obtain the monthly and annual frequencies and area burned for several Ecoregions across the WUS. We improve upon previous regression-based and ML approaches in several concrete ways; in particular, our model: relies only on the spatiotemporal variability of predictors and not on location or time based predictors (e.g., latitude or month), captures the nonlinear interactions among different predictors at multiple spatial scales, and provides robust uncertainty estimates for our results.

Our main results are as follows: *a)* the time series for both modeled frequencies and area burned are in good statistical agreement with the observed data over monthly and annual timescales at spatial scales from Ecoregions to the whole WUS; *b)* the modeled area burned successfully accounts for the interannual variability and multidecadal trends in the observed area burned in both forested and nonforested regions; *c)* for anomalous extreme fire years such as 2020, the stochastic model is useful for estimating the upper percentiles, *i.e.* $95^{\mathrm{th}}$, $99^{\mathrm{th}}$..., of the total annual area burned distribution; *d)* the cumulative
observed fire size distribution is best fit by a combined GPD model with finite mean but infinite variance, which has important consequences for how resources are allocated for fuel treatment and fire containment.

We used the SHAP technique to evaluate the predictor importance for the frequency and size models at the Ecoregional, Divisional, and WUS scales. While VPD is the leading predictor at both smaller and larger scales, the order of subleading fire month predictors – precipitation total, mean daily maximum and minimum temperatures, moisture in large diameter dead fuels
– as well as the fraction of grassland cover, aboveground biomass, and topography varies across Ecoregions, indicating that our model is able to generalize well across spatially heterogeneous climate, vegetation, and human gradients. Furthermore, we visualized the different functional relationships between predictor values and wildfire activity with potential interaction effects through partial dependence plots for several important predictors. Besides fire month variables, we find that increased fire frequencies in our model are driven by a set of antecedent predictors acting at two distinct timescales across Forests,
Deserts, and Plains: a seasonal (3-4 months) scale associated with snow or precipitation drought, and a cumulative longer term (1-2 years) scale correlated with wetter conditions that promote fuel growth. Modeled fire sizes, on the other hand, are mostly sensitive to seasonal scale antecedent conditions.

Future research directions will focus on expanding the SMLFire1.0 model framework to include: a stochastic model for human ignitions, nonstationary relationships between predictors and fire activity, fire-fuel feedback over different climate and vegetation gradients, as well as additional finer scale moisture and human action predictors. Moreover, we intend to incorporate SMLFire1.0 as a subgrid-scale parameterization scheme for the fire modules of a regional scale DGVM and ESM while also benchmarking it against existing parameterizations.

*Code and data availability.* The code for training and validating the SMLFire1.0 model as well as reproducing the figures shown in this paper is available at: https://doi.org/10.5281/zenodo.7277980. The WUMI dataset (Juang et al., 2022) contains all the fire occurrences and sizes analyzed in this paper and is hosted at: https://doi.org/10.5061/dryad.sf7m0cg72.

*Author contributions.* J.B., A.P.W., and P.G. conceived the study. A.P.W., C.J., and W.D.H. handled the data collection and processing. J.B. performed the analysis and wrote the paper. All authors discussed the results and contributed to the final version of the draft.

*Financial support.* This work is funded by the Zegar Family Foundation and the Amazon Reseach program. C.J. is supported by a Future Investigators in NASA Earth and Space Science and Technology (FINESST) Grant 80NSSC20K1617. W.D.H. acknowledges support from The Gordon and Betty Moore Foundation (GBMF10763), the Environmental Defense Fund, and the Three Cairns Group. P.G.'s research is funded by the NSF LEAP Science and Technology Center # 2019625 and the USMILE European Research Council grant.

*Competing interests.* The authors declare no competing interests.

*Acknowledgements.* We also acknowledge initial work from Kenza Amara during an internship at Columbia University on machine learning and wildfires.



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
