# Peer review of "SMLFire1.0: a stochastic machine learning (SML) model for wildfire activity in the western United States"

_EGUsphere, 2022_

## Referee Comment (RC2)

**Peer-Review of**
**"SMLFire1.0: a stochastic machine learning (SML) model for wildfire activity in the western United States"**

**Title:** SMLFire1.0: a stochastic machine learning (SML) model for wildfire activity in the western United States
**Authors:** J. Buch, A. P. Williams, C. Juang, W. D. Hansen, and P. Gentine

**Summary**

This manuscript introduces a stochastic ML fire model, SMLFire1.0, to estimate the probability distribution of monthly fire frequencies and sizes across the WUS over 1984-2020. The SMLFire1.0 model captures the interannual variability and the distinct multidecade increases in the annual area burned over different ecoregions over the WUS. The SHAP is applied to evaluate the importance of predictors and the authors find that vapor pressure deficit (VPD) is the dominant driver of fire frequencies and sizes across the WUS, followed by 1000-hour dead fuel moisture (FM1000), total monthly precipitation (Prec), mean daily maximum temperature (Tmax), and the fraction of grassland cover in a grid cell.

**Overall Feedback**

This is another work applying ML to predict fires, but this time more focusing on the WUS where fires are more connected to human activities. This take on the ML-based fire model is unique and interesting. The ML framework used here is standard, though the advantage of SMLFire1.0 compared to the gradient-boosted tree model needs more explicitly described and discussed in the main text. For instance, one of the advantages of SMLFire1.0 is uncertainty quantification. What source of uncertainty is evaluated? If it is only statistical uncertainty, would it be model-dependent? How does this model account for the spatiotemporal variability of the predictors and their non-linear interactions and why the other models couldn't? The cited work Wang et al., 2021 also considered those features in their XGBoost model and SHAP analysis. The interpolation of SHAP values also needs to be more carefully reviewed. Such as human predictors can introduce contrast impacts even at the same location. Higher population density can cause both fire ignition and suppression, a small SHAP value would overlook the impact of this predictor. For these reasons, and others mentioned below, **major revisions** to the manuscript are needed before possible publication.

**Major Remarks**

**Selection of predictors**

1. The meteorological predictors were obtained from multiple data sources. The inconsistency between the data sources may introduce additional uncertainties. Why are the daily and X-day minimum and maximum temperatures extracted from different sources? The fire weather index is calculated from relative humidity and wind speed from gridMET. Both variables from UCLA-ERA5 were used as individual predictors. Why not use the same data source to derive FFWI?

2. In addition to monthly mean daily maximum and minimum temperature, what physical information can the X-day mean variables add in? Is X-day mean calculated from the running average?

3. The predictors are selected with physical meanings. Could the authors elaborate on why a variable is chosen?

4. **Line 162**: Table S2 lists 30 predictors. This number of predictors is more close to the one after iteratively dropping off predictors that do not improve overall performance and are highly correlated as the authors mentioned in the results section. If two variables are highly correlated, which one will be kept? I would also move this to the method section.

5. Since the authors found that the antecedent precipitation is one of most the important drivers affecting plant growth, would it be helpful to include vegetation predictors that more closely connect to fuel conditions? Meanwhile, the results show low importance of the spatial variability of vegetation predictors, however, the temporal variability is important, which appears to conflict with the justification of using time-invariant biomass.

6. Surprisingly, human predictors are not among the top 10 predictors for fire frequency, while over 90 % of the California fire ignitions were associated with human activities (e.g. Balch et al., 2017).

**Explanation of SHAP values**

1. **Line 303-304** As far as I know, the Kernel SHAP also makes an assumption about feature independence. If two features are highly correlated, one value will be replaced with random ones from the background dataset, and then SHAP will generate predictions based on the new datasets while making the SHAP value estimation less reliable.

2. **Figure 11-Forests** shows increases in both grassland fraction and above-ground biomass increase the burned area. In the forests, the above-ground biomass is mainly contributed by wooden biomass. Therefore, I am wondering if grassland fraction and biomass will increase simultaneously. Could the authors plot a partial dependence plot for those two predictors? How to understand the higher $T_{max}$ would suppress fire spread? When replacing $VPD$ with relative humidity relevant variables, the relationship between $T_{max}$ and burned area becomes positive (Figure S10-lower left panel). Would the correlation between the predictors affect the results?

**Specific Remarks**

1. A stochastic ML fire model is introduced. I am wondering if any physical processes (e.g., lighting and human ignition) are included in this model?

2. Mapping the Ecoregions also on Figure 1 would much help the reader to understand their locations. The "desert" is not a place closely related to fires, considering using an alternative name for this division.

3. **Line 125**: What's the spatial resolution of ERA5-WRF? Suggest adding spatial resolution to Table S2.

4. **Line 153**: The variable Popdensity actually measures the distance to human settlements. However, the Popdensity sounds like population density which is inversely proportional to the distance. Suggest changing to Pop_dist or similar terms.

5. **Line 221-224**: While I am okay with this approximating, the MTBS dataset provides the extent of fires 1000 acres or greater in the WUS, which might be more precise than approximating each fire as a circle.

6. **Line 235-238**: Can the MDN predict burned area at each grid cell directory or for individual fires? If yes, why not sum up burned area at each grid cell or individual fires to get the Ecoregion level burned area?

7. **Line 235**: What does the spatialtemporal scale mean here? How can the CCDF plot support the breakpoint selection based on this definition?

8. **Line 242**: Should this be "consecutive time period"? Please describe how a breakpoint will be determined in the method.

9. **Line 310-315**: As stated in **Line 189**, 17489 grid cells correspond to active fires. Is it associated with the same value as stated here " ˜20000" test points?

10. **Line 361**: "are modeled".

11. **Line 451**: Which fire frequency is used in the follow on analysis?

12. **Line 449-500**: Was this statement based on all the 28 predictors or the top 10 shown in Figure 12? Increasing the distance to the camp ground does not seem to increase the burned area.

13. **Line 591-592**: The slope variables are also invariant with time.

14. **Line 603-604**: What does the "fire month" stands for?

**Reference**

Balch, Jennifer K., Bethany A. Bradley, John T. Abatzoglou, R. Chelsea Nagy, Emily J. Fusco, and Adam L. Mahood. "Human-started wildfires expand the fire niche across the United States." Proceedings of the National Academy of Sciences 114, no. 11 (2017): 2946-2951.

---

## Author Comment (AC1)

*(Note: The reviewer's comments are in gray and the author's responses are in blue. Unless specified otherwise, the line numbers quoted in our responses are with reference to the revised manuscript.)*

Overall, this study has interesting components and would be a nice contribution to the literature applying ML approach on wildfire prediction. The paper is well-written, and I think that the authors' are in a strong position to introduce the stochastic ML method, which is a relatively new concept, to the wildfire modeling community. There are a few aspects that could be further addressed before the paper is suitable for publication.

We thank the reviewer for their positive feedback on our manuscript. A detailed response to their individual comments is given below:

1a. This study used SHAP values to compare importance between predictors for the entire period as a global perspective and for each Ecoregion. But, how about the importance changes for temporal aspects (e.g. dry/wet season or extreme fire events)?

We have split the reviewer's original point into two parts for clarity. We agree that the temporal aspects of SHAP are an important diagnostic given that previous works (in particular, Wang and Wang, 2020) have shown subtle differences in the fire behavior between the dry and wet seasons in south central United States (US). Shown in Fig. 1a and Fig. 1b are the western US SHAP importance plots for the dry (May - September) and wet (October - March) seasons respectively for a MDN frequency model trained on all fires. The most significant differences between the two seasons are: the number of fires, with the dry season experiencing a factor of ~10 more fires than the wet season, and the increased importance of extreme weather variables such as $VPD^{max3}$ and $FFWI^{max3}$.

[Figure]

**Figure 1a.** SHapley Additive exPlanation (SHAP) analysis of the fire frequency MDN model outputs across the western United States for the wet (left) and dry (right) seasons. Input predictors sorted in descending order of their mean SHAP values aggregated over the entire

study period. Each colored point along the x-axis represents an individual prediction with the color corresponding to high (yellow) or low (indigo) values of the respective input predictor.

[Figure]

**Figure 1b.** Same as Figure 1a. but for the fire size MDN model.

1b.  It is interesting that none of the results in this study actually show significant importance for wind speed, although it is a key factor of fire spread.

The reviewer is correct in pointing out that wind speed is a key driver of fire spread. Indeed our results in the manuscript corroborate this fact since the monthly mean of 3-day maximum Fosberg Fire Weather Index (FFWI$^{max3}$) is an important predictor in our SHAP plots for the fire size model in almost all the ecoregions and their aggregations. The FFWI is an index of fire severity calculated using temperature, humidity, and wind speed, which has been shown to be an important correlate of wind-driven fires (Moritz et al., 2010; Barbero et al., 2014).

As such, the SHAP importance values shown in the manuscript are for a model trained with both the monthly mean of 3 day maximum wind speed (Wind) and FFWI$^{max3}$, with the FFWI$^{max3}$ being more important in all ecoregions. When we repeated our analysis after removing FFWI$^{max3}$ as a predictor, we obtained SHAP importance plots with Wind as one of the important predictors as shown in Fig. 2.

To clarify the properties of the FFWI as a correlate of hot, dry windy conditions, we have also included the following sentence in our revised manuscript (lines 124-126):

> *The FFWI, which is calculated using temperature, humidity, and wind speed (Fosberg, 1978), has been shown to be an important correlate of dry, windy conditions associated with fire weather (Moritz et al., 2010).*

[Figure]

**Figure 2.** SHapley Additive exPlanation (SHAP) analysis of the fire size MDN model outputs across the western United States with wind speed as a predictor instead of FFWI (left), and both wind speed as well as FFWI (right).

2. The discrepancy of the year 2020 (Figure 11) can be further analyzed with input predictors. Although the scale of AAB 2020 is out of the range during the training period, it can be associated with abnormal patterns in climate/vegetation or sudden changes in human induced predictors. The authors may consider this further.

The reviewer raises an interesting point that we considered while preparing our manuscript, but never explicitly addressed in its writing. Essentially, the question can be posed as follows: "Are extreme sizes for individual fires a result of extreme or anomalous values of various predictors? And since such conditions are rare, by definition, how do we train our statistical or machine learning model to correctly assign extreme responses to extreme predictor values?"

The second question is admittedly harder, but as the plots in Fig. 3 indicate: the largest fires in 2020 did *not* correspond to extreme fire weather conditions. In fact, there is a significant difference in the fire sizes of the largest fire and the fire corresponding to the most extreme value of an important predictor, potentially highlighting the role of prompt human action in containing the growth of large fires.

We consider four different types of plots to argue the above point. First, we contrast the distribution of three important fire weather predictors: VPD, FFWI$^{max3}$, and FM1000 in grid cells with and without fires. As shown in Fig. 3a, we observe a clear shift in the distributions of VPD and FM1000 in the presence and absence of fires, but there is also a sizable overlap for months with moderate fire weather. This overlap is even more significant when contrasting the predictor distributions for small and large fires in Fig. 3b. Moreover, while the most extreme predictor values correspond to large fires (red circle), the largest fire size (red diamond) occurred at moderately high but not extreme fire weather. Most strikingly, while comparing the predictor distributions for large fires that occurred between 1984-2019 and those that occurred in 2020, we find in Fig. 3c that: a) the most extreme weather conditions led to smaller fires (red, black

circles) relative to the largest fire sizes (red, black diamonds) in the respective time period; b) with the exception of FFWI$^{max3}$ values for several fires in 2020, the distributions of other predictors in grid cells with large fires were on average *less* extreme than in 1984-2019.

In summary, there appears to be only a very weak monotonic relationship between extreme predictor values and fire sizes. An important caveat being that the predictor values used in our analysis are at coarse spatial and temporal resolutions relative to the physical scales of fire front propagation. We are exploring different ways to bridge the two regimes in ongoing work.

**a)**

[Figure]

**b)**

[Figure]

**c)**

[Figure]

**Figure 3.** Contrasting probability distributions of VPD, FFWI$^{max3}$, and FM1000 values (in $\sigma$) for grid cells with: a) no fires and fires; b) small fires and large fires; c) all large fires in 1984-2019 and 2020. In b) and c), the circles indicate fires that occurred in grid cells with the most extreme predictor value, whereas the diamonds indicate the largest fire by burned area.

3. Why is 'Southness' selected rather than other directions? Also, interesting since it is included in the top 10 important predictors for the size model (Figure 12 and 13). It would be nice to further describe the role of 'Southness' in this study domain.

We appreciate the reviewer raising this comment. Mean south-facing degree of slope (also referred to as slope aspect in the fire ecology literature), or Southness, is associated with higher insolation in the Northern Hemisphere, resulting in drier conditions and low fuel moisture than all other slope directions (Rollins et al., 2002; Dillon et al., 2011). We think it is one of the strengths of our model that it is able to simultaneously assess the relative importance of climatic predictors such as VPD and topographic predictors like Slope and Southess simultaneously.

We included the following sentence (lines 161-163) in the Data section of our revised manuscript to clarify the reviewer's comment:

> *In the Northern Hemisphere, Southness is associated with higher insolation which results in drier conditions and low fuel moisture relative to other slope directions (Rollins et al., 2002; Dillon et al., 2011).*

4. A typo in L361 : 'are modeled'

We have fixed this typo in the revised manuscript.

**References:**

1.  Barbero, R., Abatzoglou, J. T., Steel, E. A., & Larkin, N. K. (2014). Modeling very large-fire occurrences over the continental United States from weather and climate forcing. Environmental Research Letters, 9(12), 124009, doi:10.1088/1748-9326/9/12/124009

2.  Moritz, M. A., Moody, T. J., Krawchuk, M. A., Hughes, M., and Hall, A. (2010), Spatial variation in extreme winds predicts large wildfire locations in chaparral ecosystems, Geophys. Res. Lett., 37, L04801, doi:10.1029/2009GL041735

3.  Dillon, G. K., Z. A. Holden, P. Morgan, M. A. Crimmins, E. K. Heyerdahl, and C. H. Luce. (2011). Both topography and climate affected forest and woodland burn severity in two regions of the western US, 1984 to 2006. Ecosphere 2(12):130, doi:10.1890/ES11-00271.1

4.  Wang, S. S.-C., & Wang, Y. (2020). Quantifying the effects of environmental factors on wildfire burned area in the south central US using integrated machine learning techniques. *Atmospheric Chemistry and Physics*, *20*(18), 11065–11087. https://doi.org/10.5194/acp-20-11065-2020

---

## Author Comment (AC2)

*(Note: The reviewer's comments are in gray and the author's responses are in blue. Unless specified otherwise, the line numbers quoted in our responses are with reference to the revised manuscript.)*

Jatan Buch et al. developed a SMLFire model based on Mixture Density Networks and monthly climate, land surface, and atmospheric conditions. This work focused on both fire frequency and total burnt area over Western US. In general, this study is timely and important. The high performance of SMLFire is exciting for both fire frequency and burnt area. The presentation is smooth and well-done. Congratulations. Below are my comments and recommendations.

We thank the reviewer for their positive feedback on our manuscript. A detailed response to their individual comments is given below:

1. Fuel load seems missing in the input variable list, which is an important predictor for fire spread thus burnt area. Also GDP (missing) is often considered as an important indicator of human effects on fire management and firefighting efforts. Some others are also potentially useful to consider, e.g. road density. I would suggest creating a new table with a full list of input variables and explaining how these variables possibly affect fire frequency and burnt area.

We agree with the reviewer's comments that fuel load is an important input variable for fire activity. Ideally, we would like to include dynamic fuel load variables that track changes in fuel density from climate perturbations as well as previous fires. However, we could not find any fuel load products solely based on observations in the literature, so we used the fractional land cover outputs from the National Land Cover Database (NLCD) as input predictors instead. We have also mentioned (lines 148-150 in the original manuscript) that a promising future direction of research is to precisely accomplish what the reviewer suggests: include fuel load variables such as dead and live biomass from a dynamic vegetation model.

Since our fire model is based only on the western US, it is unclear to us how GDP, typically a national level economic indicator, will be a helpful predictor. We would appreciate helpful references from the reviewer on this point.

We have now included the following table as Table S3 in our supplementary information section as per the reviewer's suggestion to clarify the qualitative effect of individual variables on fire frequency and burned area:

| Predictors | Qualitative effect | | Comments |
|---|---|---|---|
| | Fire frequency | Fire size | |
| VPD, AntVPD_$M$mon, VPD$^{maxX}$ | ↑ | ↑ | VPD on multiple timescales, from weekly to seasonal, is positively correlated with both fire frequency and size. |
| Tmax, | ↑ | ↑ | Tmax on multiple timescales, from weekly to |

| | | | |
|---|---|---|---|
| AntTmax_*M*mon Tmax$^{maxX}$ | | | seasonal, is positively correlated with both fire frequency and size. |
| Tmin, Tmin$^{maxX}$ | / | ↑ | Both extreme Tmin and monthly mean Tmin are positively correlated with fire size. Tmin is not a significant predictor for fire frequency. |
| Prec, AntPrec_*M*mon | ↓ | ↓ | Prec on multiple timescales, from monthly to seasonal, is negatively correlated with both fire frequency and size. |
| AntPrec_lag1, AntPrec_lag2 | ↑ | ↑ | Annual mean of Prec in lagging years, a proxy for biomass growth, is positively correlated with fire frequency and size. |
| SWE_mean, SWE_max AvgSWE_*M*mon | ↓ | ↓ | Snow water equivalent on multiple timescales, from monthly to seasonal, is negatively correlated with both fire frequency and size. |
| FM1000 | ↓ | ↓ | 1000-hour dead fuel moisture is negatively correlated with fire frequency and size. |
| FFWI, FFWI$^{maxX}$ | ↑ | ↑ | Mean and extreme values of FFWI are positively correlated with fire frequency and size. |
| Wind$^{maxX}$ | / | ↑ | Monthly maxima of X-day mean wind speed is positively correlated with fire size. Wind speed is not a significant predictor for fire frequency. |
| Biomass | ↕ | ↑ | Spatial variance in biomass is positively correlated with fire size, however its effect on fire frequency is ambiguous with potential confounding by human action predictors. |
| Grassland, Shrubland | ↑ | ↑ | Fraction of grassland and shrubland cover increases fuel flammability and continuity over a landscape, and is thus positively correlated with fire frequency and fire size. |
| Lightning | ↑ | / | Increased lightning strike density contributes additional ignitions, and is positively correlated with fire frequency. Lightning is not a significant predictor of fire size. |
| Slope | ↑ | ↑ | Slope is positively correlated with fire frequency and size since the rate of fire spread is proportional to the degree of slope. |
| Southness | ↑ | ↑ | Southness, or mean south-facing degree of slope, dictates the level of solar insolation and is positively correlated with fire frequency and size. |

| Pop10_dist | ↕ | ↑ | Increased distance from areas with population density greater than 10 km$^2$ is a correlate of remoteness leading to a larger fire size. Its effect on fire frequency is ambiguous since these areas experience fewer ignitions while also having reduced access to early fire containment efforts. |
|---|---|---|---|

**Table 1.** Summary of the qualitative effect and physical meaning of important model predictors on fire frequency and burned area. The symbols ↑, ↓, ↕, and / refer to positive, negative, ambiguous, and insignificant correlations between a predictor and fire response variable respectively.

2. Spatial evaluation of SMLFire simulation is limited to regions, but evaluation on gridcell scale is also important because the model is gridcell-based and spatially-explicit. Suggest showing spatial maps of simulated vs observed western US fire frequency and burnt area statistics for long-term mean, decadal trend etc. It is interesting to see 12-km scale spatial hot spots of trends and variability as well.

Thank you for raising an important point. We think spatial validation of our model at the 12-km scale is difficult for two main reasons:

i) Most 12 km grid cells (≳ 99%) do not experience any fires in the study period of ~40 years, and only ~5% of the grid cells with fires experience more than 1 fire in any month. Even on decadal scales, fires are incredibly rare, so unless the model is trained on 1-2 degree (~50-100 km) grid size scale, it will not contain enough fires for a meaningful trend. Thus, we choose aggregate EPA Level III (L3) ecoregions to demonstrate the monthly, interannual, and decadal variability of fire frequency and burned area across the western United States.

ii) Moreover, the stochasticity of monthly scale climate predictors that our model is trained on also contributes to the lack of spatial precision over longer timescales. Most papers in the literature that simulate long-term trends in fire probability (for example, Parisien and Moritz, 2009; Chen et. al., 2021) end up relying on climate normals (*i.e.* long-term averages) as predictors. We believe that one of the strengths of our models is its ability to leverage this stochasticity for projecting a range of possible outcomes, which is more useful than accuracy for planning fire mitigation.

3. Human vs natural ignited fires have clear differences in ignition location and background climate. I understand that SMLFire does not distinguish human vs natural fire, but it is worth exploring or discussion on how that might bias SMLFire in simulating spatial-temporal distribution of fires as well as the interpretation of underlying control factors for fire frequency and burnt area.

We appreciate the reviewer's comment about accounting for the difference in human vs natural ignitions. As a preliminary exploration of potential biases in SMLFire, we performed the following experiment: based on Fig. 1 of Balch et al., 2017, we identified that a large proportion of human-started fires ($\gtrsim$ 60% of all fires in that ecoregion) in our western United States (WUS) study region occur in Mediterranean California (CA) ecoregions as well as coastal parts of the Pacific Northwest (PNW). Next, we trained SMLFire on fires from all 5 L3 ecoregions from this area and performed the SHAP analysis for fire frequency predictors.

In Fig. 1 below, we compared the SHAP values for all fires in the CA+PNW ecoregions from the above experiment with the results for SMLFire trained on fires across the WUS (as shown in Fig. 6 of the manuscript, which is also reproduced as the right panel of Fig. 1). We find that relative to the WUS case, the SHAP values for Lightning are more important for the CA+PNW case. However, unlike the WUS case, we find that Campnum, or the mean number of campsites in a grid cell, emerges as an important predictor of fire frequency. None of the other human related predictors are selected in our experiment.

[Figure]

**Figure 1.** Mean absolute SHAP values for all fires in the CA+PNW ecoregions with the frequency MDN model trained on: (Left) only fires from the 5 L3 ecoregions in the CA+PNW area; (Right) fires from across the WUS as shown in Fig. 6 of the manuscript.

Based on the experiment outlined above, we have added the following sentences to our discussion clarifying the potential bias in SMLFire while modeling fires with human vs natural ignitions:

(lines 421-423) *Given that a large fraction of fires in parts of the WUS, especially Mediterranean California and coastal PNW (Balch et. al. 2017), are human ignited, this result could stem from a skewed sampling of fires while training SMLFire1.0 as well as the lack of correlation between our chosen human predictors and fire occurrences.*

(lines 557-559) *Alternatively, we could leverage the seasonal differences between human and lightning started fires to account for potential selection biases in training data for SMLFire1.0.*

4. It's not clear how uncertainty quantification is done. Does it only consider parametric uncertainty? How about model structure (how many layers of hidden layer, number of neurons for each layer), other hyperparameters? How about forcing data uncertainties?

The mixture density network used in our analysis learns the function map between input predictors and the parameters of a mixture distribution for each individual fire. This may be interpreted as approximating the likelihood of fire occurrence or size given the input predictors. We estimate the parametric model uncertainty by performing Monte Carlo simulations on a frequency or size MDN model with parameters fixed to their optimal values and calculating the variance of the samples.

Our framework does not account for uncertainties due to hyperparameter or forcing data uncertainties in this analysis. However, given that we are approximating the likelihood function, we can easily embed it in a hierarchical Bayesian model to account for the hyperparameter and data forcing uncertainties. Typically, we expect the data forcing uncertainties to be a much bigger factor while using SMLFire in conjunction with seasonal and subseasonal-to-seasonal (S2S) climate model forecasts.

We have edited the following sentence in the Methods section to clarify the reviewer's point (lines 223-224):

> *We treat the variance as an estimate of the parametric model uncertainty, or equivalently the uncertainty in modeled frequency due to different realizations of a parametric model.*

We have also lightly edited the first paragraph of the Conclusions section (lines 603-608) to make the point about parametric model uncertainty estimation clearer.

**References:**

1. Parisien, M. and Moritz, M.A. (2009), Environmental controls on the distribution of wildfire at multiple spatial scales. Ecological Monographs, 79: 127-154. https://doi.org/10.1890/07-1289.1

2. Chen, B., Jin, Y., Scaduto, E., Moritz, M. A., Goulden, M. L., & Randerson, J. T. (2021). Climate, fuel, and land use shaped the spatial pattern of wildfire in California's Sierra Nevada. *Journal of Geophysical Research: Biogeosciences*, 126, e2020JG005786. https://doi.org/10.1029/2020JG005786

3. Balch, J. K., Bradley, B. A., Abatzoglou, J. T., Nagy, R. C., Fusco, E. J., & Mahood, A. L. (2017). Human-started wildfires expand the fire niche across the United States. Proceedings of the National Academy of Sciences, 114(11), 2946–2951. https://doi.org/10.1073/pnas.1617394114

---

## Author Comment (AC3)

*(Note: The reviewer's comments are in gray and the author's responses are in blue. Unless specified otherwise, the line numbers quoted in our responses are with reference to the revised manuscript.)*

**Overall Feedback**

This is another work applying ML to predict fires, but this time more focusing on the WUS where fires are more connected to human activities. This take on the ML-based fire model is unique and interesting. The ML framework used here is standard, though the advantage of SMLFire1.0 compared to the gradient-boosted tree model needs to be more explicitly described and discussed in the main text. For instance, one of the advantages of SMLFire1.0 is uncertainty quantification. What source of uncertainty is evaluated? If it is only statistical uncertainty, would it be model-dependent? How does this model account for the spatiotemporal variability of the predictors and their non-linear interactions and why the other models couldn't? The cited work Wang et al., 2021 also considered those features in their XGBoost model and SHAP analysis. The interpolation of SHAP values also needs to be more carefully reviewed. Such as human predictors can introduce contrast impacts even at the same location. Higher population density can cause both fire ignition and suppression, a small SHAP value would overlook the impact of this predictor. For these reasons, and others mentioned below, major revisions to the manuscript are needed before possible publication.

We thank the reviewer for carefully reading through our manuscript and providing constructive feedback on our ML framework as well as its interpretation through SHAP values.

One major point raised by the reviewer is regarding the comparison of SMLFire1.0 with the gradient-boosted tree model of Wang et. al., 2021. While the Wang et. al., 2021 model also accounts for the spatiotemporal variability of predictors and their nonlinear interactions, we identify two major differences, namely their gradient-boosted tree model: a) only predicts the area burned at the grid cell level and not the fire probability or frequency, thus missing an important element of stochasticity in fire activity; b) does not perform uncertainty quantification, while our model uses the variance of Monte Carlo samples from the optimized mixture model to estimate the parametric model uncertainty for fire frequency and sizes. Although we view the analysis of Wang et. al., 2021 as complementary to ours, we make the distinction between our approaches sharper in the Theory section as follows:

(lines 184-191)

> *c) be based on parametric distributions that could be sampled using Monte Carlo simulations for estimating the mean and parametric model uncertainty of modeled fire frequency and sizes. While tree-based ML approaches using xGBoost have shown high performance in area burned prediction across the continental US (Wang et al., 2021), we adopt a neural network based architecture here because it combines the flexibility of machine learning techniques with the robustness of parametric distribution based methods traditionally used in statistical fire modeling (Westerling et al., 2011, Joseph et al., 2019). Moreover, since neural network models have more powerful representation*

*learning capabilities than gradient-boosted trees (Levin et al., 2022), they are better equipped for generalizing the learned relationships between input predictors and fires to test data from future climate states or different fire regimes.*

(lines 227-228):

*We treat the variance as an estimate of the parametric model uncertainty, or equivalently the uncertainty in modeled frequency due to different realizations of a parametric model.*

The first paragraph of the Conclusions section (lines 608-616) has also been lightly edited to make the point about parametric model uncertainty estimation clearer.

We address the reviewer's comments regarding the interpretation of SHAP values and possible confounders in the section on "Explanation of SHAP values" below.

**Major Remarks**

**Selection of predictors**

1. The meteorological predictors were obtained from multiple data sources. The inconsistency between the data sources may introduce additional uncertainties. Why are the daily and X-day minimum and maximum temperatures extracted from different sources? The fire weather index is calculated from relative humidity and wind speed from gridMET. Both variables from UCLA-ERA5 were used as individual predictors. Why not use the same data source to derive FFWI?

We thank the reviewer for raising these points. Firstly, we used the NOAA nClimgrid data to obtain the monthly means of daily maximum and minimum temperature data, which is only available on monthly (and not daily) scales. We preferred the NOAA data over gridMET because it is of higher quality and extends back to 1895, which will enable us to validate our model on fire data from the early 20th century in forthcoming work.

In order to calculate the X-day daily maximum and minimum temperatures we used daily scale data from the UCLA-ERA5 reanalysis. More importantly, we used temperature, humidity, and wind speed data from UCLA-ERA5 reanalysis to calculate the monthly mean FFWI and X-day maximum FFWI. We acknowledge an unfortunate error in the original manuscript (line 124) where we stated that FFWI data was taken from gridMET. The only predictor obtained from gridMET in our analysis is the monthly mean FM1000 value.

We could have used daily scale data from gridMET for all the above predictors besides FM1000, however since wind speed in gridMET is derived by downscaling NARR data from a coarser resolution of 32 km x 32 km resolution, we instead used wind speeds from UCLA-ERA5 reanalysis which downscaled ERA5 wind data to a higher 9 km x 9 km resolution using the WRF model. Figure 11 in Rahimi et. al., 2022 illustrates the improvement in wind speed resolution due to UCLA ERA5-WRF as compared to gridMET-NARR. In order to ensure minimal error from

using multiple data sources as the reviewer has pointed out, all X-day maximum predictors as well as monthly mean FFWI have been derived using data from UCLA ERA5-WRF reanalysis.

Following is the revised data description in the manuscript (lines 123-129):

> *Monthly mean FM1000 values, an indicator of climate-derived moisture balance, were adapted from gridMET (Abatzoglou, 2013).The FFWI, which is calculated using temperature, humidity, and wind speed (Fosberg, 1978), has been shown to be an important correlate of dry, windy conditions associated with fire weather (Moritz et al., 2010). Since wind speed in gridMET is derived using a spatial interpolation of the National Atmospheric Regional Reanalysis (NARR) data from a coarser (32 km x 32 km) resolution, we instead use high (9 km x 9 km) resolution temperature, humidity, and wind speed predictors from the dynamically downscaled UCLA ERA5-WRF reanalysis (Rahimi et al., 2022) to calculate the monthly mean FFWI.*

2. In addition to monthly mean daily maximum and minimum temperature, what physical information can the X-day mean variables add in? Is X-day mean calculated from the running average?

We have expanded the discussion around the current description of X-day mean variables and also commented upon the physical meaning behind their inclusion (lines 129-132):

> *Furthermore, we use daily scale data from the UCLA ERA5-WRF reanalysis to calculate the monthly maximum X-day running average of daily maximum and minimum temperature ($Tmax^{maxX}$, $Tmin^{maxX}$), where $X \in \{3, 5, 7\}$. Similar X-day extreme predictors are also derived for VPD, FFWI, and wind speed. The X-day running average of these predictors are included to improve our model's sensitivity to weekly scale extreme fire weather caused by events such as heatwaves.*

3. The predictors are selected with physical meanings. Could the authors elaborate on why a variable is chosen?

Besides adding an explanation for including FM1000, FFWI, and X-day extreme weather variables in the Data section, we have now included an additional table in the Supplementary information, Table S3, that elaborates briefly on the relationship between each predictor and fire response variables.

4. Line 162: Table S2 lists 30 predictors. This number of predictors is more close to the one after iteratively dropping off predictors that do not improve overall performance and are highly correlated as the authors mentioned in the results section. If two variables are highly correlated, which one will be kept? I would also move this to the method section.

We apologize for the confusion regarding the number of variables in Table S2. We have added an extra line in the caption explaining how the total number of potential predictors in Table S2 adds up to 51:

*Considering each predictor's M antecedent months' average and maximum X-day running average components as distinct predictors, the total number of predictors adds up to 51.*

As per the reviewer's suggestion, we have also moved the discussion regarding variable selection to the Methods section (lines 316-320).

5. Since the authors found that the antecedent precipitation is one of the most important drivers affecting plant growth, would it be helpful to include vegetation predictors that more closely connect to fuel conditions? Meanwhile, the results show low importance of the spatial variability of vegetation predictors, however, the temporal variability is important, which appears to conflict with the justification of using time-invariant biomass.

[Figure]

**Figure 1.** Spatial map of the correlation between the mean of antecedent precipitation in the previous year and aboveground biomass aggregated for each Level III ecoregion. The boundaries of various ecoregions are indicated by solid black lines.

The reviewer raises an important point regarding the importance of temporal variability of vegetation predictors. We acknowledge that the abundance and flammability of fine fuels is one of the main drivers of fire activity. Time-varying biomass maps would be an ideal predictor for our model; however, to the best of our knowledge, no such products exist in the fire ecology or remote sensing literature.

Thus, we use a combination of both: a static biomass map to inform the model of relative fuel abundance as well as antecedent precipitation predictions in lag years as a proxy of dynamic plant growth. Interestingly, as we show in Fig. 1 above, the long term mean of $AntPrec_{lag1}$ shows moderate correlations with the time-invariant aboveground biomass across different L3 Ecoregions.

We also note from Fig. S2, S3 and S7, S8 in our Supplementary Information that $AntPrec_{lag1}$ is an important predictor for fire frequency while Biomass is an important predictor for fire size. A potential explanation could be that while $AntPrec_{lag1}$ is responsible for plant growth that drives fire frequency in arid climates, the spatial fuel abundance given by Biomass, especially in Forests, is a major driver of fire spread resulting in larger burned areas.

6. Surprisingly, human predictors are not among the top 10 predictors for fire frequency, while over 90% of the California fire ignitions were associated with human activities (e.g. Balch et al., 2017).

This is an insightful comment and is related to point 3 of Referee Comment (RC) #3 on our original preprint. Before outlining how we addressed the reviewer's comment, we note that the FPA-FOD data used in Balch et. al., 2017 contains significantly more smaller fires than the WUMI dataset that we are using which only contains fire ≥ 1 $km^2$. Since human fires are mostly small fires, we expect the ~90% number for human started fires in California to be slightly lower for the WUMI dataset.

There are two potential explanations for the lack of human predictors among top 10 predictors for fire frequency: a) since our model is trained on fires across the WUS where the proportion of human started fires is ≤ 50%, there could be a potential skewed sampling of fires while training SMLFire1.0 (note that we do not have access to labels for human vs natural ignitions in our data set); b) our choices of human predictors are poor correlated with fire occurrences.

As outlined in RC#3, we also performed a toy experiment to test the explanations outlined above. Based on our findings, we have added the following sentences to our discussion clarifying the potential bias in SMLFire while modeling fires with human vs natural ignitions:

(lines 431-434) *Given that a large fraction of fires in parts of the WUS, especially Mediterranean California and coastal PNW (Balch et. al. 2017), are human ignited, this result could stem from a skewed sampling of fires while training SMLFire1.0 as well as the lack of correlation between our chosen human predictors and fire occurrences.*

(lines 568-570) *Alternatively, we could leverage the seasonal differences between human and lightning started fires to account for potential selection biases in training data for SMLFire1.0.*

**Explanation of SHAP values**

1. Line 303-304 As far as I know, the Kernel SHAP also makes an assumption about feature independence. If two features are highly correlated, one value will be replaced with random

ones from the background dataset, and then SHAP will generate predictions based on the new datasets while making the SHAP value estimation less reliable.

We agree with the reviewer's comment that the Kernel SHAP method assumes feature independence while calculating the Shapley additive value of a particular feature and also relies on random draws from the dataset for replacing absent features. Thus, in case there are correlated features, Kernel SHAP could end up overweighting the importance of unlikely data points. We have modified our writing in the 'Predictor Importance' subsection to clarify this point (lines 325-329):

> *This is in contrast to the traditional predictor importance techniques which only rely on a fixed coalition of predictors to assess the contribution of an individual variable … We note that a drawback of using the Kernel Explainer method is that its assumption of predictor independence could lead, in practice, to a biased estimation of predictor importance in the presence of two or more strongly correlated features.*

2. Figure 11-Forests shows increases in both grassland fraction and above-ground biomass increase the burned area. In the forests, the above-ground biomass is mainly contributed by wooden biomass. Therefore, I am wondering if grassland fraction and biomass will increase simultaneously. Could the authors plot a partial dependence plot for those two predictors? How to understand the higher Tmax would suppress fire spread? When replacing VPD with relative humidity relevant variables, the relationship between Tmax and burned area becomes positive (Figure S10-lower left panel). Would the correlation between the predictors affect the results?

We find that the Ecoregions are the more appropriate spatial scale to assess the relative predictor importance of Biomass and Grassland. Analyzing Figs. S7 and S8, it is clear that the cumulative importance of Grassland at the Forests Division level is primarily driven by its role in CA Central and South Coasts, Southern Rockies, and AZ/NM Mountains Ecoregions, whereas Biomass is the more important predictor in Sierra Nevada, CA North Coast, PNW Mountains, and Northern Rockies. For completeness, we also include a plot comparing the partial dependence plot for Biomass and Grassland for the aggregated Forests Division:

[Figure]

Indeed, as the reviewer's previous comment suggests, correlation between VPD and Tmax appears to affect the predictor importance of Tmax in Forests (Fig. 13) relative to the case with only RH and Tmax (Fig. S10; lower left panel). We have now added an additional line in our Results section clarifying this point (lines 542-545):

> *From the perspective of predictor importance, there might actually be an advantage to using RH instead of VPD: the correlation between VPD and Tmax leads to a small but spurious trend in SHAP value for Tmax in Forests as shown in Fig. 13, whereas using only RH and Tmax in Fig. S10 yields the correct Tmax effect on fire size. On the other hand, from the perspective of future climate-fire relationships, …*

**Specific Remarks**

1. A stochastic ML fire model is introduced. I am wondering if any physical processes (e.g., lighting and human ignition) are included in this model?

We have included lightning the lightning strike density as a correlate of natural ignitions and several human predictors for human ignitions. However, unlike process-based models such as SPITFIRE (Thonicke et. al., 2010), we do not include any physical parameterizations for ignitions. We appreciate the reviewer's suggestion here and will consider including additional physics in SMLFire1.0 to improve its ignition modeling capacity.

We have also included two additional sentences in the Data section to clarify the inclusion of human predictors and the dual role played by predictors such as population density (lines 166-168):

> *These predictors serve as potential correlates of human ignitions for fire occurrences as well as proxies for access to fire suppression or containment resources. Some predictors such as Popdensity could play a dual role through both increasing the likelihood of ignitions while also providing easier access for fire suppression.*

2. Mapping the Ecoregions also on Figure 1 would much help the reader to understand their locations. The "desert" is not a place closely related to fires, considering using an alternative name for this division.

We acknowledge the scope for confusion due to our use of "desert" to describe the arid grassland and shrubland regions spanning several Ecoregions through the center of our study region. However, we chose "desert" following the nomenclature used by the EPA while referring to the Level II ecoregion, North American Cold and Warm Deserts, containing the area roughly equivalent to our Deserts Division.

3. Line 125: What's the spatial resolution of ERA5-WRF? Suggest adding spatial resolution to Table S2.

We have added spatial resolution to Table S2 as an additional column, and also included a clarification in the caption about how all predictors, despite having different native resolutions, are aggregated to the 12 km resolution in our statistical analysis.

4. Line 153: The variable Popdensity actually measures the distance to human settlements. However, the Popdensity sounds like population density which is inversely proportional to the distance. Suggest changing to Pop_dist or similar terms.

We have modified all references to 'Popdensity' throughout the text as well as within plots to 'Pop10_dist' following the reviewer's suggestion.

5. Line 221-224: While I am okay with this approximating, the MTBS dataset provides the extent of fires 1000 acres or greater in the WUS, which might be more precise than approximating each fire as a circle.

We agree with the reviewer's comment; the main consideration for approximating the fire shape as a circle was driven by the absence of burned area polygons in the WUMI dataset, which contains both interagency and MTBS fires. We have added the following sentence (lines 241-242) in the Methods section to clarify the reviewer's comment:

> *In future work, we will use burned area polygons from MTBS for large fires instead of the circular approximation while deriving the effective input predictors.*

6. Line 235-238: Can the MDN predict burned area at each grid cell directory or for individual fires? If yes, why not sum up burned area at each grid cell or individual fires to get the Ecoregion level burned area?

We derive the Ecoregion level burned area by summing up the areas for individual fires – exactly as the reviewer has suggested (see lines 249-250 in the revised manuscript). In the referenced passage, we have outlined a simple analytic expression for estimating the relative contribution from frequency as well as fire sizes to the mean burned area at a given scale. For instance, if we were to use a ML model for fire frequency and overpredict the number of fires but underpredict individual fire sizes, we would still get the correct burned area. Using the analytic expression provides a helpful diagnostic tool in such a scenario.

We have lightly edited our writing around this point in the revised manuscript to emphasize that the burned area is calculated by summing up individual fires, and the average burned area calculation is only for interpreting our calculation schematically.

7. Line 235: What does the spatialtemporal scale mean here? How can the CCDF plot support the breakpoint selection based on this definition?

As we described above, the use of spatiotemporal scale here is merely schematic. The breakpoint selection procedure is based on creating the CCDF plot with individual fire sizes and not cumulative burned area.

8. Line 242: Should this be "consecutive time period"? Please describe how a breakpoint will be determined in the method.

We have outlined our procedure for determining the breakpoint year as well as different validation steps in the Results section (lines 456 - 465). A similar version of the text can be found in the preprint (lines 435-444) but may have been obscured due to the unfortunate page break due to the placement of Figs. 9 and 10.

9. Line 310-315: As stated in Line 189, 17489 grid cells correspond to active fires. Is it associated with the same value as stated here "˜20000" test points?

To construct the set of test points for our SHAP values, we consider all (i.e 17,489) grid cells with a fire and combine them in a ratio of 1:3 with a random sample of background points with no fires (i.e 17,489/3 ~ 5830), bringing the total number of test points to 23,319 points, or ~20,000 as we mention in the text.

10. Line 361: "are modeled".

We have fixed this typo in the revised manuscript.

11. Line 451: Which fire frequency is used in the follow on analysis?

Both modeled and observed fire frequency are used to calculate the monthly and annual area burned across the WUS. Another source of stochasticity that we explore in Fig. S6 is deriving the burned area using input predictors corresponding to observed as well as modeled fire locations.

12. Line 449-500: Was this statement based on all the 28 predictors or the top 10 shown in Figure 12? Increasing the distance to the camp ground does not seem to increase the burned area.

We are slightly confused by the reviewer's comments here since we do not discuss the effect of distance to camp grounds in Fig. 12. The one human related variable that shows up as an important predictor in Fig. 12 is the distance to areas with population density greater than 10 people per square kilometer (Popdensity; or Pop10_dist in the revised manuscript), which indicates that more remote areas are more conducive to larger fire sizes.

13. Line 591-592: The slope variables are also invariant with time.

This is a good point. We have amended lines 591-592 (now lines 613-615) as follows:

> …our model: relies only on the spatiotemporal variability of dynamic predictors, the spatial variability of static predictors, and not on any predictors related to the location and time such as latitude or calendar month;...

14. Line 603-604: What does the "fire month" stand for?

We have adopted "fire month" as a descriptor throughout the manuscript to refer to monthly scale predictors for any month with a fire. We mainly use it to distinguish the effect of climate and fire weather conditions during a month from the influence of antecedent climate predictors. We have now clarified this usage in the Data section by modifying the following sentence (line 130) where we first use the term fire month:

*Thus, for a given month m with potential fire activity (henceforth fire month) …*

**References:**

1. Balch, J. K., Bradley, B. A., Abatzoglou, J. T., Nagy, R. C., Fusco, E. J., & Mahood, A. L. (2017). Human-started wildfires expand the fire niche across the United States. Proceedings of the National Academy of Sciences, 114(11), 2946–2951. https://doi.org/10.1073/pnas.1617394114

2. Moritz, M. A., Moody, T. J., Krawchuk, M. A., Hughes, M., and Hall, A. (2010), Spatial variation in extreme winds predicts large wildfire locations in chaparral ecosystems, Geophys. Res. Lett., 37, L04801, doi:10.1029/2009GL041735

3. Rahimi, S., Krantz, W., Lin, Y.-H., Bass, B., Goldenson, N., Hall, A., et al. (2022). Evaluation of a reanalysis-driven configuration of WRF4 over the western United States from 1980 to 2020. *Journal of Geophysical Research: Atmospheres*, 127, e2021JD035699. https://doi.org/10.1029/2021JD035699

4. Thonicke, K., Spessa, A., Prentice, I. C., Harrison, S. P., Dong, L., & Carmona-Moreno, C. (2010). The influence of vegetation, fire spread and fire behaviour on biomass burning and trace gas emissions: Results from a process-based model. *Biogeosciences*, *7*(6), 1991–2011. https://doi.org/10.5194/bg-7-1991-2010

---

## Referee Report (RR1)

**Peer-Review of**
**"SMLFire1.0: a stochastic machine learning (SML) model for wildfire activity in the western United States"**

**Title:** SMLFire1.0: a stochastic machine learning (SML) model for wildfire activity in the western United States
**Authors:** J. Buch, A. P. Williams, C. Juang, W. D. Hansen, and P. Gentine
**Reviewed by** Ye Liu, ye.liu@pnnl.gov

**Summary**

I would like to express my sincere gratitude to the authors for their efforts in fully addressing my comments and concerns regarding the manuscript. The revised version of the paper now includes substantial documentation of the novelty, particularly in comparison to the previous machine learning-based fire prediction approach. The authors have provided clear and compelling explanations for their predictor selection process, integrating important physical information into their approach. Furthermore, the consistent and thorough explanation of SHAP values is appreciated, as well as the potential impact of including correlated predictors in the model. Overall, the revisions have significantly improved the quality and impact of the manuscript. I wish to thank the authors for the opportunity to review their interesting work. It is a nice contribution to the field of climate-fire interaction. I just have a few minor technical comments that I hope the author can address before publication. Thus, my recommendation to GMD is to publish the work pending these technical corrections.

**Minor Comments**

1. **Line 7-8**: Is this relationship based on observation?

2. **Figure 1**: What test method and any threshold are used to define statically significant?

3. **Line 158**: Suggest change "urban" to "urban fraction" if the fraction is used.

4. **Line 159**: Suggest change to "..., Pop10_dist defined as ..."

---

## Author Response (AR3)

*(Note: The reviewers' comments are in gray and the author's responses are in blue. Unless specified otherwise, the line numbers quoted in our responses are with reference to the revised manuscript.)*

**Reviewer #1**

Overall, this study has interesting components and would be a nice contribution to the literature applying ML approach on wildfire prediction. The paper is well-written, and I think that the authors' are in a strong position to introduce the stochastic ML method, which is a relatively new concept, to the wildfire modeling community. There are a few aspects that could be further addressed before the paper is suitable for publication.

We thank the reviewer for their positive feedback on our manuscript. A detailed response to their individual comments is given below:

1a. This study used SHAP values to compare importance between predictors for the entire period as a global perspective and for each Ecoregion. But, how about the importance changes for temporal aspects (e.g. dry/wet season or extreme fire events)?

We have split the reviewer's original point into two parts for clarity. We agree that the temporal aspects of SHAP are an important diagnostic given that previous works (in particular, Wang and Wang, 2020) have shown subtle differences in the fire behavior between the dry and wet seasons in south central United States (US). Shown in Figs. R1a and R1b are the western US SHAP importance plots for the dry (May - September) and wet (October - March) seasons respectively for a MDN frequency model trained on all fires. The most significant differences between the two seasons are: the number of fires, with the dry season experiencing a factor of ~10 more fires than the wet season, and the increased importance of extreme weather variables such as $VPD^{max3}$ and $FFWI^{max3}$.

[Figure]

**Figure R1a.** SHapley Additive exPlanation (SHAP) analysis of the fire frequency MDN model outputs across the western United States for the wet (left) and dry (right) seasons. Input

predictors sorted in descending order of their mean SHAP values aggregated over the entire study period. Each colored point along the x-axis represents an individual prediction with the color corresponding to high (yellow) or low (indigo) values of the respective input predictor.

[Figure]

**Figure R1b.** Same as Fig. R1a but for the fire size MDN model.

1b. It is interesting that none of the results in this study actually show significant importance for wind speed, although it is a key factor of fire spread.

The reviewer is correct in pointing out that wind speed is a key driver of fire spread. Indeed our results in the manuscript corroborate this fact since the monthly mean of 3-day maximum Fosberg Fire Weather Index (FFWI$^{max3}$) is an important predictor in our SHAP plots for the fire size model in almost all the ecoregions and their aggregations. The FFWI is an index of fire severity calculated using temperature, humidity, and wind speed, which has been shown to be an important correlate of wind-driven fires (Moritz et al., 2010; Barbero et al., 2014).

As such, the SHAP importance values shown in the manuscript are for a model trained with both the monthly mean of 3 day maximum wind speed (Wind) and FFWI$^{max3}$, with the FFWI$^{max3}$ being more important in all ecoregions. When we repeated our analysis after removing FFWI$^{max3}$ as a predictor, we obtained SHAP importance plots with Wind as one of the important predictors as shown in Fig. R2, as anticipated by the reviewer.

To clarify the properties of the FFWI as a correlate of hot, dry windy conditions, we have also included the following sentence in our revised manuscript,

L124-125: *The FFWI, which is calculated using temperature, humidity, and wind speed (Fosberg, 1978), has been shown to be an important correlate of dry, windy conditions associated with fire weather (Moritz et al., 2010).*

[Figure]

**Figure R2.** SHapley Additive exPlanation (SHAP) analysis of the fire size MDN model outputs across the western United States with wind speed as a predictor instead of FFWI (left), and both wind speed as well as FFWI (right).

2. The discrepancy of the year 2020 (Figure 11) can be further analyzed with input predictors. Although the scale of AAB 2020 is out of the range during the training period, it can be associated with abnormal patterns in climate/vegetation or sudden changes in human induced predictors. The authors may consider this further.

The reviewer raises an interesting point that we considered while preparing our manuscript, but never explicitly addressed in its writing. Essentially, the question can be posed as follows: "Are extreme sizes for individual fires a result of extreme or anomalous values of various predictors?"

As the plots in Fig. R3 indicate: individually, the largest fires in 2020 across the WUS did *not* correspond to extreme fire weather conditions at either monthly or sub-monthly scales. In fact, there is a significant difference in the fire sizes of the largest fire and the fire corresponding to the most extreme value of an important predictor, potentially highlighting the confounding role of prompt human action in containing the growth of large fires.

We consider three different plots to argue the above point. First, we contrast the distribution of three important fire weather predictors: VPD, FFWI$^{max3}$, and FM1000 in grid cells with and without fires. As shown in Fig. R3a, we observe a clear shift in the distributions of VPD and FM1000 in the presence and absence of fires, but there is also a sizable overlap for months with moderate fire weather. This overlap is even more significant when contrasting the predictor distributions for small and large fires in Fig. R3b. Moreover, while the most extreme predictor values correspond to large fires (red circle), the largest fire size (red diamond) occurred at moderately high but not extreme fire weather on both monthly and sub-monthly scales. Most strikingly, while comparing the predictor distributions for large fires that occurred between 1984-2019 and those that occurred in 2020, we find in Fig. R3c that: a) the most extreme weather conditions led to smaller fires (red, black circles) relative to the largest fire sizes (red, black diamonds) in the respective time period; b) with the exception of FFWI$^{max3}$ values for

several fires in 2020, the distributions of other predictors in grid cells with large fires were on average *less* extreme than in 1984-2019.

In summary, there appears to be only a weak monotonic relationship between extreme predictor values and individual fire sizes. An important caveat being that the predictor values used in our analysis are at coarse spatial and temporal resolutions relative to the physical scales of fire front propagation. We are exploring different ways to bridge the two regimes in ongoing work. Following are the changes in the text that summarize this point,

L571-574: *For the fire size model, a major limitation of our current approach is its reliance on climate predictors whose spatial and temporal scales are coarse relative to the physical scales involved in fire front propagation (Bakhshaii and Johnson, 2019). Bridging these two regimes (Sullivan, 2009) is an important focus of our ongoing work to improve predictability of extreme fire behavior.*

**a)**

[Figure]

**b)**

[Figure]

**c)**

[Figure]

**Figure R3.** Contrasting probability distributions of VPD, FFWI$^{max3}$, and FM1000 values (in $\sigma$) for grid cells with: a) no fires and fires; b) small fires and large fires; c) all large fires in 1984-2019 and 2020. In b) and c), the circles indicate fires that occurred in grid cells with the most extreme predictor value, whereas the diamonds indicate the largest fire by burned area.

3. Why is 'Southness' selected rather than other directions? Also, interesting since it is included in the top 10 important predictors for the size model (Figure 12 and 13). It would be nice to further describe the role of 'Southness' in this study domain.

We appreciate the reviewer raising this comment. Mean south-facing degree of slope (also referred to as slope aspect in the fire ecology literature), or Southness, is associated with higher insolation in the Northern Hemisphere, resulting in drier conditions and low fuel moisture than all other slope directions (Rollins et al., 2002; Dillon et al., 2011). We think it is one of the strengths of our model that it is able to simultaneously assess the relative importance of climatic predictors such as VPD and topographic predictors like Slope and Southess simultaneously.

We have included the following sentence in the Data section of our revised manuscript to clarify the reviewer's comment,

L170-172: *In the Northern Hemisphere, Southness is associated with higher insolation which results in drier conditions and low fuel moisture relative to other slope directions (Rollins et al., 2002; Dillon et al., 2011).*

4. A typo in L361 : 'are modeled'

We have fixed this typo in the revised manuscript.

**References:**

1.  Barbero, R., Abatzoglou, J. T., Steel, E. A., & Larkin, N. K. (2014). Modeling very large-fire occurrences over the continental United States from weather and climate forcing. Environmental Research Letters, 9(12), 124009, doi:10.1088/1748-9326/9/12/124009

2.  Moritz, M. A., Moody, T. J., Krawchuk, M. A., Hughes, M., and Hall, A. (2010), Spatial variation in extreme winds predicts large wildfire locations in chaparral ecosystems, Geophys. Res. Lett., 37,

L04801, doi:10.1029/2009GL041735

3. Dillon, G. K., Z. A. Holden, P. Morgan, M. A. Crimmins, E. K. Heyerdahl, and C. H. Luce. (2011). Both topography and climate affected forest and woodland burn severity in two regions of the western US, 1984 to 2006. Ecosphere 2(12):130, doi:10.1890/ES11-00271.1

4. Wang, S. S.-C., & Wang, Y. (2020). Quantifying the effects of environmental factors on wildfire burned area in the south central US using integrated machine learning techniques. *Atmospheric Chemistry and Physics*, *20*(18), 11065–11087. https://doi.org/10.5194/acp-20-11065-2020

**Reviewer #2**

**Overall Feedback**

This is another work applying ML to predict fires, but this time more focusing on the WUS where fires are more connected to human activities. This take on the ML-based fire model is unique and interesting. The ML framework used here is standard, though the advantage of SMLFire1.0 compared to the gradient-boosted tree model needs to be more explicitly described and discussed in the main text. For instance, one of the advantages of SMLFire1.0 is uncertainty quantification. What source of uncertainty is evaluated? If it is only statistical uncertainty, would it be model-dependent? How does this model account for the spatiotemporal variability of the predictors and their non-linear interactions and why the other models couldn't? The cited work Wang et al., 2021 also considered those features in their XGBoost model and SHAP analysis. The interpolation of SHAP values also needs to be more carefully reviewed. Such as human predictors can introduce contrast impacts even at the same location. Higher population density can cause both fire ignition and suppression, a small SHAP value would overlook the impact of this predictor. For these reasons, and others mentioned below, major revisions to the manuscript are needed before possible publication.

We thank the reviewer for carefully reading through our manuscript and providing constructive feedback on our ML framework as well as its interpretation through SHAP values.

One major point raised by the reviewer is regarding the comparison of SMLFire1.0 with the gradient-boosted tree model of Wang et. al., (2021). While the Wang et. al., (2021) model also accounts for the spatiotemporal variability of predictors and their nonlinear interactions, we identify two major differences, namely their gradient-boosted tree model: a) only predicts the area burned at the grid cell level and not the fire probability or frequency, thus missing an important element of stochasticity in fire activity; b) does not perform uncertainty quantification, while our model uses the variance of Monte Carlo samples from the optimized mixture model to estimate the parametric model uncertainty for fire frequency and sizes. Although we view the analysis of Wang et. al., (2021) as complementary to ours, we make the following distinction between our approaches sharper in the Theory section,

L184-191: *c) be based on parametric distributions that could be sampled using Monte Carlo simulations for estimating the mean and parametric model uncertainty of modeled fire frequency and sizes. While tree-based ML approaches using xGBoost have shown high performance in area burned prediction across the continental US (Wang et al., 2021), we adopt a neural network based architecture here because it combines the flexibility of machine learning techniques with the robustness of parametric distribution based methods traditionally used in statistical fire modeling (Westerling et al., 2011, Joseph et al., 2019). Additionally, since neural network models are more powerful at learning feature representations than gradient-boosted trees (Levin et al., 2022), they are better equipped for generalizing the learned relationships between input predictors and fires to test data from future climate states or different fire regimes.*

L227-228: *We treat the variance as an estimate of the parametric model uncertainty, or equivalently the uncertainty in modeled frequency due to different realizations of a parametric model.*

The first paragraph of the Conclusions section (L608-616) has also been lightly edited to make the point about parametric model uncertainty estimation clearer.

We address the reviewer's comments regarding the interpretation of SHAP values and possible confounders in the section on "Explanation of SHAP values" below.

**Major Remarks**

**Selection of predictors**

1. The meteorological predictors were obtained from multiple data sources. The inconsistency between the data sources may introduce additional uncertainties. Why are the daily and X-day minimum and maximum temperatures extracted from different sources? The fire weather index is calculated from relative humidity and wind speed from gridMET. Both variables from UCLA-ERA5 were used as individual predictors. Why not use the same data source to derive FFWI?

We thank the reviewer for raising these points. Firstly, we used the NOAA nClimgrid data to obtain the monthly means of daily maximum and minimum temperature data, which is only available on monthly (and not daily) scales. We preferred the NOAA data over gridMET because it is of higher quality and extends back to 1895, which will enable us to validate our model on fire data from the early 20th century in forthcoming work.

In order to calculate the X-day daily maximum and minimum temperatures we used daily scale data from the UCLA-ERA5 reanalysis. More importantly, we used temperature, humidity, and wind speed data from UCLA-ERA5 reanalysis to calculate the monthly mean FFWI and X-day maximum FFWI. We acknowledge an unfortunate error in the original manuscript (line 124) where we stated that FFWI data was taken from gridMET. The only predictor obtained from gridMET in our analysis is the monthly mean FM1000 value.

We could have used daily scale data from gridMET for all the above predictors besides FM1000, however since wind speed in gridMET is derived by downscaling NARR data from a coarser resolution of 32 km x 32 km resolution, we instead used wind speeds from UCLA-ERA5 reanalysis which downscaled ERA5 wind data to a higher 9 km x 9 km resolution using the WRF regional climate model. Figure 11 in Rahimi et. al., (2022) illustrates the improvement in wind speed resolution due to UCLA ERA5-WRF as compared to gridMET-NARR. Following is the revised data description in the manuscript,

L123-129: *Monthly mean FM1000 values, an indicator of climate-derived moisture balance, were adapted from gridMET (Abatzoglou, 2013).The FFWI, which is calculated using temperature, humidity, and wind speed (Fosberg, 1978), has been shown to be an important correlate of dry, windy conditions associated with fire weather (Moritz et al., 2010). Since wind speed in gridMET is derived using a spatial interpolation of the National Atmospheric Regional Reanalysis (NARR) data from a coarser (32 km x 32 km) resolution, we instead use high (9 km x 9 km) resolution temperature, humidity, and wind speed predictors from the dynamically*

*downscaled UCLA ERA5-WRF reanalysis (Rahimi et al., 2022) to calculate the monthly mean FFWI.*

2. In addition to monthly mean daily maximum and minimum temperature, what physical information can the X-day mean variables add in? Is X-day mean calculated from the running average?

We have expanded the discussion around the current description of X-day mean variables and also commented upon the physical meaning behind their inclusion,

L129-132: *Furthermore, we use daily scale data from the UCLA ERA5-WRF reanalysis to calculate the monthly maximum X-day running average of daily maximum and minimum temperature ($Tmax^{maxX}$, $Tmin^{maxX}$), where X ∈ {3, 5, 7}. Similar X-day extreme predictors are also derived for VPD, FFWI, and wind speed. The X-day running average of these predictors are included to improve our model's sensitivity to weekly scale extreme fire weather caused by events such as heatwaves.*

3. The predictors are selected with physical meanings. Could the authors elaborate on why a variable is chosen?

Besides adding an explanation for including FM1000, FFWI, and X-day extreme weather variables in the Data section, we have now included an additional table in the Supplementary information, Table S3, that elaborates briefly on the relationship between each predictor and fire response variables.

4. Line 162: Table S2 lists 30 predictors. This number of predictors is more close to the one after iteratively dropping off predictors that do not improve overall performance and are highly correlated as the authors mentioned in the results section. If two variables are highly correlated, which one will be kept? I would also move this to the method section.

We have added an extra line in the caption explaining how the total number of potential predictors in Table S2 adds up to 51:

*Considering each predictor's M antecedent months' average and maximum X-day running average components as distinct predictors, the total number of predictors adds up to 51.*

As per the reviewer's suggestion, we have also moved the discussion regarding variable selection to the Methods section (L316-320).

5. Since the authors found that the antecedent precipitation is one of the most important drivers affecting plant growth, would it be helpful to include vegetation predictors that more closely connect to fuel conditions? Meanwhile, the results show low importance of the spatial variability of vegetation predictors, however, the temporal variability is important, which appears to conflict with the justification of using time-invariant biomass.

[Figure]

**Figure R1.** Spatial map of the correlation between the mean of antecedent precipitation in the previous year (AntPrec$_{lag1}$) and aboveground biomass aggregated for each Level III ecoregion. The boundaries of various ecoregions are indicated by solid black lines.

The reviewer raises an important point regarding the importance of temporal variability of vegetation predictors. We acknowledge that the abundance and flammability of fine fuels is one of the main drivers of fire activity. Time-varying biomass maps would be an ideal predictor for our model; however, to the best of our knowledge, no such products exist in the fire ecology or remote sensing literature.

Thus, we use a combination of both: a static biomass map to inform the model of relative fuel abundance as well as antecedent precipitation predictions in lag years as a proxy of dynamic plant growth. Interestingly, as we show in Fig. R1 above, the long term mean of AntPrec$_{lag1}$ shows moderate correlations with the time-invariant aboveground biomass across different L3 Ecoregions.

We also note from Fig. S2, S3 and S7, S8 in our Supplementary Information that AntPrec$_{lag1}$ is an important predictor for fire frequency while Biomass is an important predictor for fire size. A potential explanation could be that while AntPrec$_{lag1}$ is responsible for plant growth that drives fire frequency in arid climates, the spatial fuel abundance given by Biomass, especially in Forests, is a major driver of fire spread resulting in larger burned areas.

6. Surprisingly, human predictors are not among the top 10 predictors for fire frequency, while over 90% of the California fire ignitions were associated with human activities (e.g. Balch et al., 2017).

This is an insightful comment and is related to point 3 of Referee Comment (RC) #3 on our original preprint. Before outlining how we addressed the reviewer's comment, we note that the FPA-FOD data used in Balch et. al., 2017 contains significantly more smaller fires than the WUMI dataset that we are using which only contains fire $\geq 1$ km$^2$. Since human fires are mostly small fires, we expect the ~90% number for human started fires in California to be slightly lower for the WUMI dataset.

There are two potential explanations for the lack of human predictors among top 10 predictors for fire frequency: a) since our model is trained on fires across the WUS where the proportion of human started fires is $\leq 50\%$, there could be a potential skewed sampling of fires while training SMLFire1.0 (note that we do not have access to labels for human vs natural ignitions in our data set); b) our choices of human predictors are poorly correlated with fire occurrences.

As outlined in RC#3, we also performed a toy experiment to test the explanations outlined above. Based on our findings, we have added the following sentences to our discussion clarifying the potential bias in SMLFire while modeling fires with human vs natural ignitions:

L431-434:*Given that a large fraction of fires in parts of the WUS, especially Mediterranean California and coastal PNW are human ignited (Balch et. al. 2017), this result could stem from a skewed sampling of fires while training SMLFire1.0 as well as the lack of correlation between our chosen human predictors and fire occurrences.*

L568-570: *Alternatively, we could leverage the seasonal differences between human and lightning started fires to account for potential selection biases in training data for SMLFire1.0.*

**Explanation of SHAP values**

1. Line 303-304 As far as I know, the Kernel SHAP also makes an assumption about feature independence. If two features are highly correlated, one value will be replaced with random ones from the background dataset, and then SHAP will generate predictions based on the new datasets while making the SHAP value estimation less reliable.

We agree with the reviewer's comment that the Kernel SHAP method assumes feature independence while calculating the Shapley additive value of a particular feature and also relies on random draws from the dataset for replacing absent features. Thus, in case there are correlated features, Kernel SHAP could end up overweighting the importance of unlikely data points. We have modified our writing in the 'Predictor Importance' subsection to clarify this point,

L325-329: *This is in contrast to the traditional predictor importance techniques which only rely on a fixed coalition of predictors to assess the contribution of an individual variable … We note that a drawback of using the Kernel Explainer method is that its assumption of predictor*

*independence could lead, in practice, to a biased estimation of predictor importance in the presence of two or more strongly correlated features.*

2. Figure 11-Forests shows increases in both grassland fraction and above-ground biomass increase the burned area. In the forests, the above-ground biomass is mainly contributed by wooden biomass. Therefore, I am wondering if grassland fraction and biomass will increase simultaneously. Could the authors plot a partial dependence plot for those two predictors? How to understand the higher Tmax would suppress fire spread? When replacing VPD with relative humidity relevant variables, the relationship between Tmax and burned area becomes positive (Figure S10-lower left panel). Would the correlation between the predictors affect the results?

We find that Ecoregions are the more appropriate spatial scale to assess the relative predictor importance of Biomass and Grassland. Analyzing Figs. S7 and S8, it is clear that the cumulative importance of Grassland at the Forests Division level is primarily driven by its role in CA Central and South Coasts, Southern Rockies, and AZ/NM Mountains Ecoregions, whereas Biomass is the more important predictor in Sierra Nevada, CA North Coast, PNW Mountains, and Northern Rockies. For completeness, we also show in Fig. R2 a comparison between the partial dependence plots for Biomass and Grassland for the aggregated Forests Division:

[Figure]

**Figure R2.** SHAP dependence plots for Biomass (left) and Grassland (right) predictors of the MDN size model.

Indeed, as the reviewer's previous comment suggests, correlation between VPD and Tmax appears to affect the predictor importance of Tmax in Forests (Fig. 13) relative to the case with only RH and Tmax (Fig. S10; lower left panel). We have now added an additional line in our Results section clarifying this point,

L542-547: *From the perspective of predictor importance, there might actually be an advantage to using RH instead of VPD: the correlation between VPD and Tmax leads to a small but spurious trend in SHAP value for Tmax in Forests as shown in Fig. 13, whereas using only RH and Tmax in Fig. S10 yields the correct Tmax effect on fire size. On the other hand, from the*

*perspective of future climate-fire relationships, the fact that the decision of including VPD, RH, or both does not substantially affect model performance does not mean that this decision is unimportant.*

**Specific Remarks**

1. A stochastic ML fire model is introduced. I am wondering if any physical processes (e.g., lighting and human ignition) are included in this model?

We have included lightning the lightning strike density as a correlate of natural ignitions and several human predictors for human ignitions. However, unlike process-based models such as SPITFIRE (Thonicke et. al., 2010), we do not include any physical parameterizations for ignitions. We appreciate the reviewer's comment here and we will consider including additional physics in future versions of SMLFire1.0 to improve its ignition modeling capacity.

We have also included two additional sentences in the Data section to clarify the inclusion of human predictors and the dual role played by predictors such as population density,

L166-168: *These predictors serve as potential correlates of human ignitions for fire occurrences as well as proxies for access to fire suppression or containment resources. Some predictors such as Popdensity could play a dual role through both increasing the likelihood of ignitions while also providing easier access for fire suppression.*

2. Mapping the Ecoregions also on Figure 1 would much help the reader to understand their locations. The "desert" is not a place closely related to fires, considering using an alternative name for this division.

We acknowledge the scope for confusion due to our use of "Desert" to describe the arid grassland and shrubland regions spanning several Ecoregions through the center of our study region. However, we chose "Desert" following the nomenclature used by the EPA while referring to the Level II ecoregion, North American Cold and Warm Deserts, containing the area roughly equivalent to our Deserts Division.

3. Line 125: What's the spatial resolution of ERA5-WRF? Suggest adding spatial resolution to Table S2.

We have added spatial resolution to Table S2 as an additional column, and also included a clarification in the caption about how all predictors, despite having different native resolutions, are aggregated to the 12 km resolution in our statistical analysis.

4. Line 153: The variable Popdensity actually measures the distance to human settlements. However, the Popdensity sounds like population density which is inversely proportional to the distance. Suggest changing to Pop_dist or similar terms.

We have modified all references to 'Popdensity' throughout the text as well as within plots to 'Pop10_dist' following the reviewer's suggestion.

5. Line 221-224: While I am okay with this approximating, the MTBS dataset provides the extent of fires 1000 acres or greater in the WUS, which might be more precise than approximating each fire as a circle.

We agree with the reviewer's comment; the main consideration for approximating the fire shape as a circle was driven by the absence of burned area polygons in the WUMI dataset, which contains both interagency and MTBS fires. We have added the following sentence in the Methods section to clarify the reviewer's comment.

L241-242: *In future work, we will use burned area polygons from MTBS for large fires instead of the circular approximation while deriving the effective input predictors.*

6. Line 235-238: Can the MDN predict burned area at each grid cell directory or for individual fires? If yes, why not sum up burned area at each grid cell or individual fires to get the Ecoregion level burned area?

We derive the Ecoregion level burned area by summing up the areas for individual fires – exactly as the reviewer has suggested (see lines 249-250 in the revised manuscript). In the referenced passage, we have outlined a simple analytic expression for estimating the relative contribution from frequency as well as fire sizes to the mean burned area at a given scale. For instance, if we were to use a ML model for fire frequency and overpredict the number of fires but underpredict individual fire sizes, we would still get the correct burned area. Using the analytic expression provides a helpful diagnostic tool in such a scenario.

We have lightly edited our writing around this point in the revised manuscript to emphasize that the burned area is calculated by summing up individual fires, and the average burned area calculation is only for interpreting our calculation schematically.

7. Line 235: What does the spatialtemporal scale mean here? How can the CCDF plot support the breakpoint selection based on this definition?

As we described above, the use of spatiotemporal scale here is merely schematic. The breakpoint selection procedure is based on creating the CCDF plot with individual fire sizes and not cumulative burned area.

8. Line 242: Should this be "consecutive time period"? Please describe how a breakpoint will be determined in the method.

We have outlined our procedure for determining the breakpoint year as well as different validation steps in the Results section (lines 456 - 465). A similar version of the text can be found in the original manuscript (lines 435-444) but may have been obscured due to the unfortunate page break due to the placement of Figs. 9 and 10.

9. Line 310-315: As stated in Line 189, 17489 grid cells correspond to active fires. Is it associated with the same value as stated here "~20000" test points?

To construct the set of test points for our SHAP values, we consider all (i.e 17,489) grid cells with a fire and combine them in a ratio of 1:3 with a random sample of background points with no fires (i.e 17,489/3 ~ 5830), bringing the total number of test points to 23,319 points, or ~20,000 as we mention in the text.

10. Line 361: "are modeled".

We have fixed this typo in the revised manuscript.

11. Line 451: Which fire frequency is used in the follow on analysis?

Both modeled and observed fire frequency are used to calculate the monthly and annual area burned across the WUS. Another source of stochasticity that we explore in Fig. S6 is deriving the burned area using input predictors corresponding to observed as well as modeled fire locations.

12. Line 449-500: Was this statement based on all the 28 predictors or the top 10 shown in Figure 12? Increasing the distance to the camp ground does not seem to increase the burned area.

We are slightly confused by the reviewer's comments here since we do not discuss the effect of distance to camp grounds in Fig. 12. The one human related variable that shows up as an important predictor in Fig. 12 is the distance to areas with population density greater than 10 people per square kilometer (Popdensity; or Pop10_dist in the revised manuscript), which indicates that more remote areas are more conducive to larger fire sizes.

13. Line 591-592: The slope variables are also invariant with time.

This is a good point. We have amended lines 591-592 (now lines 616-618) as follows:

> …our model: relies only on the spatiotemporal variability of dynamic predictors, the spatial variability of static predictors, and not on any predictors related to the location and time such as latitude or calendar month;...

14. Line 603-604: What does the "fire month" stand for?

We have adopted "fire month" as a descriptor throughout the manuscript to refer to monthly scale predictors for any month with a fire. We mainly use it to distinguish the effect of climate and fire weather conditions during a month from the influence of antecedent climate predictors. We have now clarified this usage in the Data section by modifying the following sentence where we first use the term fire month,

 L136: *Thus, for a given month m with potential fire activity (henceforth fire month) …*

**References:**

1. Balch, J. K., Bradley, B. A., Abatzoglou, J. T., Nagy, R. C., Fusco, E. J., & Mahood, A. L. (2017). Human-started wildfires expand the fire niche across the United States. Proceedings of the National Academy of Sciences, 114(11), 2946–2951. https://doi.org/10.1073/pnas.1617394114

2. Moritz, M. A., Moody, T. J., Krawchuk, M. A., Hughes, M., and Hall, A. (2010), Spatial variation in extreme winds predicts large wildfire locations in chaparral ecosystems, Geophys. Res. Lett., 37, L04801, doi:10.1029/2009GL041735

3. Rahimi, S., Krantz, W., Lin, Y.-H., Bass, B., Goldenson, N., Hall, A., et al. (2022). Evaluation of a reanalysis-driven configuration of WRF4 over the western United States from 1980 to 2020. *Journal of Geophysical Research: Atmospheres*, 127, e2021JD035699. https://doi.org/10.1029/2021JD035699

4. Thonicke, K., Spessa, A., Prentice, I. C., Harrison, S. P., Dong, L., & Carmona-Moreno, C. (2010). The influence of vegetation, fire spread and fire behaviour on biomass burning and trace gas emissions: Results from a process-based model. *Biogeosciences*, 7(6), 1991–2011. https://doi.org/10.5194/bg-7-1991-2010

**Reviewer #3**

Jatan Buch et al. developed a SMLFire model based on Mixture Density Networks and monthly climate, land surface, and atmospheric conditions. This work focused on both fire frequency and total burnt area over Western US. In general, this study is timely and important. The high

performance of SMLFire is exciting for both fire frequency and burnt area. The presentation is smooth and well-done. Congratulations. Below are my comments and recommendations.

We thank the reviewer for their positive feedback on our manuscript. A detailed response to their individual comments is given below:

1. Fuel load seems missing in the input variable list, which is an important predictor for fire spread thus burnt area. Also GDP (missing) is often considered as an important indicator of human effects on fire management and firefighting efforts. Some others are also potentially useful to consider, e.g. road density. I would suggest creating a new table with a full list of input variables and explaining how these variables possibly affect fire frequency and burnt area.

We agree with the reviewer's comments that fuel load is an important input variable for fire activity. Ideally, we would like to include dynamic fuel load variables that track changes in fuel density from climate perturbations as well as previous fires. However, we could not find any fuel load products solely based on observations in the literature, so we used the fractional land cover outputs from the National Land Cover Database (NLCD) as input predictors instead. We have also mentioned (lines 148-150 in the original manuscript) that a promising future direction of research is to precisely accomplish what the reviewer suggests: include fuel load variables such as dead and live biomass from a dynamic vegetation model.

We appreciate the reviewer's suggestion of including GDP as a predictor for our model. Unfortunately, one challenge of using socioeconomic predictors, such as GDP, is that they are often only available at the county or census block level which is much coarser than the 12 km x 12 km grid cells we are using in our analysis. In the future, gridded socioeconomic predictors will be a powerful addition. It is worth noting however, that the majority of resources used to suppress fire are allocated at the national level, not the county level.

We have now included the following table (Table R1) as Table S3 in our supplementary information section as per the reviewer's suggestion to clarify the qualitative effect of individual variables on fire frequency and burned area:

| Predictors | Qualitative effect | | Comments |
| --- | --- | --- | --- |
| | Fire frequency | Fire size | |
| VPD, AntVPD_$M$mon, VPD$^{maxX}$ | ↑ | ↑ | VPD on multiple timescales, from weekly to seasonal, is positively correlated with both fire frequency and size. |
| Tmax, AntTmax_$M$mon Tmax$^{maxX}$ | ↑ | ↑ | Tmax on multiple timescales, from weekly to seasonal, is positively correlated with both fire frequency and size. |
| Tmin, | / | ↑ | Both extreme Tmin and monthly mean Tmin are |

| | | | |
|---|---|---|---|
| Tmin$^{maxX}$ | | | positively correlated with fire size. Tmin is not a significant predictor for fire frequency. |
| Prec, AntPrec_$M$mon | ↓ | ↓ | Prec on multiple timescales, from monthly to seasonal, is negatively correlated with both fire frequency and size. |
| AntPrec_lag1, AntPrec_lag2 | ↑ | ↑ | Annual mean of Prec in lagging years, a proxy for biomass growth, is positively correlated with fire frequency and size. |
| SWE_mean, SWE_max AvgSWE_$M$mon | ↓ | ↓ | Snow water equivalent on multiple timescales, from monthly to seasonal, is negatively correlated with both fire frequency and size. |
| FM1000 | ↓ | ↓ | 1000-hour dead fuel moisture is negatively correlated with fire frequency and size. |
| FFWI, FFWI$^{maxX}$ | ↑ | ↑ | Mean and extreme values of FFWI are positively correlated with fire frequency and size. |
| Wind$^{maxX}$ | / | ↑ | Monthly maxima of X-day mean wind speed is positively correlated with fire size. Wind speed is not a significant predictor for fire frequency. |
| Biomass | ↕ | ↑ | Spatial variance in biomass is positively correlated with fire size, however its effect on fire frequency is ambiguous with potential confounding by human action predictors. |
| Grassland, Shrubland | ↑ | ↑ | Fraction of grassland and shrubland cover increases fuel flammability and continuity over a landscape, and is thus positively correlated with fire frequency and fire size. |
| Lightning | ↑ | / | Increased lightning strike density contributes additional ignitions, and is positively correlated with fire frequency. Lightning is not a significant predictor of fire size. |
| Slope | ↑ | ↑ | Slope is positively correlated with fire frequency and size since the rate of fire spread is proportional to the degree of slope. |
| Southness | ↑ | ↑ | Southness, or mean south-facing degree of slope, dictates the level of solar insolation and is positively correlated with fire frequency and size. |
| Pop10_dist | ↕ | ↑ | Increased distance from areas with population density greater than 10 km$^2$ is a correlate of remoteness leading to a larger fire size. Its effect |

| | | | on fire frequency is ambiguous since these areas experience fewer ignitions while also having reduced access to early fire containment efforts. |
|---|---|---|---|
| | | | |

**Table R1.** Summary of the physical meaning of important model predictors as well as their qualitative effect on fire frequency and size. The symbols ↑, ↓, ↕, and / refer to positive, negative, ambiguous, and insignificant correlations between a predictor and fire response variable respectively.

2. Spatial evaluation of SMLFire simulation is limited to regions, but evaluation on gridcell scale is also important because the model is gridcell-based and spatially-explicit. Suggest showing spatial maps of simulated vs observed western US fire frequency and burnt area statistics for long-term mean, decadal trend etc. It is interesting to see 12-km scale spatial hot spots of trends and variability as well.

Thank you for raising an important point. We think spatial validation of our model at the 12-km scale is difficult for two main reasons:

i) Most 12 km grid cells (≳ 99%) do not experience any fires in the study period of ~40 years, and only ~5% of the grid cells with fires experience more than 1 fire in any month. Even on decadal scales, fires are incredibly rare, so unless the model is trained on 1-2 degree (~50-100 km) grid size scale, it will not contain enough fires for a meaningful trend. Thus, we choose aggregate EPA Level III (L3) ecoregions to demonstrate the monthly, interannual, and decadal variability of fire frequency and burned area across the western United States.

ii) Moreover, the stochasticity of monthly scale climate predictors that our model is trained on also contributes to the lack of spatial precision over longer timescales. Most papers in the literature that simulate long-term trends in fire probability (for example, Parisien and Moritz, 2009; Chen et. al., 2021) end up relying on climate normals (*i.e.* long-term averages) as predictors. We believe that one of the strengths of our models is its ability to leverage this stochasticity for projecting a range of possible outcomes, which is more useful than accuracy for planning fire mitigation.

3. Human vs natural ignited fires have clear differences in ignition location and background climate. I understand that SMLFire does not distinguish human vs natural fire, but it is worth exploring or discussion on how that might bias SMLFire in simulating spatial-temporal distribution of fires as well as the interpretation of underlying control factors for fire frequency and burnt area.

We appreciate the reviewer's comment about accounting for the difference in human vs natural ignitions. As a preliminary exploration of potential biases in SMLFire, we performed the following experiment: based on Fig. 1 of Balch et al., 2017, we identified that a large proportion of human-started fires (≳ 60% of all fires in that ecoregion) in our western United States (WUS)

study region occur in Mediterranean California (CA) ecoregions as well as coastal parts of the Pacific Northwest (PNW). Next, we trained SMLFire on fires from all 5 L3 ecoregions from this area and performed the SHAP analysis for fire frequency predictors.

In Fig. R1 below, we compared the SHAP values for all fires in the CA+PNW ecoregions from the above experiment with the results for SMLFire trained on fires across the WUS (as shown in Fig. 6 of the manuscript, which is also reproduced in the right panel of Fig. R1). We find that relative to the WUS case, the SHAP values for Lightning are more important for the CA+PNW case. However, unlike the WUS case, we find that Campnum, or the mean number of campsites in a grid cell, emerges as an important predictor of fire frequency. None of the other human related predictors are selected in our experiment.

[Figure]

**Figure R1.** Mean absolute SHAP values for all fires in the CA+PNW ecoregions with the frequency MDN model trained on: (Left) only fires from the 5 L3 ecoregions in the CA+PNW area; (Right) fires from across the WUS as shown in Fig. 6 of the manuscript.

Based on the experiment outlined above, we have added the following sentences to our discussion clarifying the potential bias in SMLFire while modeling fires with human vs natural ignitions:

L431-434: *Given that a large fraction of fires in parts of the WUS, especially Mediterranean California and coastal PNW are human ignited (Balch et. al. 2017), this result could stem from a skewed sampling of fires while training SMLFire1.0 as well as the lack of correlation between our chosen human predictors and fire occurrences.*

L568-570: *Alternatively, we could leverage the seasonal differences between human and lightning started fires to account for potential selection biases in training data for SMLFire1.0.*

4. It's not clear how uncertainty quantification is done. Does it only consider parametric uncertainty? How about model structure (how many layers of hidden layer, number of neurons for each layer), other hyperparameters? How about forcing data uncertainties?

The mixture density network used in our analysis learns the function map between input predictors and the parameters of a mixture distribution for each individual fire. This may be interpreted as approximating the likelihood of fire occurrence or size given the input predictors. We estimate the parametric model uncertainty by performing Monte Carlo simulations on a frequency or size MDN model with parameters fixed to their optimal values and calculating the variance of the samples.

Our framework does not account for uncertainties due to hyperparameter or forcing data uncertainties in this analysis. However, given that we are approximating the likelihood function, a promising direction for future work is to embed the likelihood in a hierarchical Bayesian model to account for the hyperparameter and data forcing uncertainties. Typically, we expect the data forcing uncertainties to be a much bigger factor while using SMLFire in conjunction with seasonal and subseasonal-to-seasonal (S2S) climate model forecasts. We have edited the following sentence in the Methods section to clarify the reviewer's point,

L227-228: *We treat the variance as an estimate of the parametric model uncertainty, or equivalently the uncertainty in modeled frequency due to different realizations of a parametric model.*

We have also lightly edited the first paragraph of the Conclusions section (lines 614-615) to make the point about parametric model uncertainty estimation clearer.

**References:**

1. Parisien, M. and Moritz, M.A. (2009), Environmental controls on the distribution of wildfire at multiple spatial scales. Ecological Monographs, 79: 127-154. https://doi.org/10.1890/07-1289.1

2. Chen, B., Jin, Y., Scaduto, E., Moritz, M. A., Goulden, M. L., & Randerson, J. T. (2021). Climate, fuel, and land use shaped the spatial pattern of wildfire in California's Sierra Nevada. *Journal of Geophysical Research: Biogeosciences*, 126, e2020JG005786. https://doi.org/10.1029/2020JG005786

3. Balch, J. K., Bradley, B. A., Abatzoglou, J. T., Nagy, R. C., Fusco, E. J., & Mahood, A. L. (2017). Human-started wildfires expand the fire niche across the United States. Proceedings of the National Academy of Sciences, 114(11), 2946–2951. https://doi.org/10.1073/pnas.1617394114

**Response to request for minor revision**

Dear Editor,

We thank both reviewers for their positive feedback on our manuscript. A detailed response to their individual comments is provided below:

Line 7-8: Is this relationship based on observation?

We assume that this comment was regarding the correlation between annual time series of both fire variables at the ecoregion level. The relevant sentence in our abstract has now been amended to clarify that the correlation is between the **modeled** time series and **observations**.

L7-8: *Moreover, the modeled annual time series of both fire variables exhibit strong correlations (r ≥ 0.6) with observations in 16 out of 18 ecoregions.*

Figure 1: What test method and any threshold are used to define statically significant?

We used the Student's t-test to determine statistically significant trends (i.e rejecting the null hypothesis that there is no trend) with the p-value threshold, $p < 0.05$. We have now mentioned the threshold clearly in the Figure 1 caption as well as included the following line in the main text,

L111-112: *We also indicate all statistically significant ($p < 0.05$) trends, which were determined using the Student's t-test.*

Line 158: Suggest change "urban" to "urban fraction" if the fraction is used.

We have changed the predictor name from "Urban" to "Urban fraction" (L159)

Line 159: Suggest change to "..., Pop10_dist defined as ..."

We have improved the writing around population predictors (L161-162) as suggested by the reviewer. Now, the Pop10_dist line reads: "*distance from the nearest area with population density greater than 10 people per square kilometer (Pop10_dist), …*"

Line 190: I disagree with the notion that neural network models are always better than gradient boosting models or other ML models, particularly when it comes to inferring out-of-sample climates such as future climate states or different fire regimes as mentioned by the authors. The limitations of NN for future inference were previously discussed on [1,2]. I recommend to modify the sentence.

We have now modified the sentence in L190 to reflect the reviewer's comment about highlighting the limitations of ML models.

L190-193: *Recent work (Levin et al., 2022) has also shown that neural network models are more powerful at learning feature representations than gradient-boosted trees. However, generalizing the learned relationships between input predictors and fires to out-of-sample data from future climate states or different fire regimes remains a challenging problem for most ML*

*approaches, including neural network based models (Rasp et al., 2018; Yuval and O'Gorman, 2020).*

Figure 4, 5, 8-10: it is hard to compare lines in the monthly scale evaluation. It would be nice to adjust line thickness or the plot design to facilitate comparison.

To facilitate the comparison between observations and modeled fire frequency and burned area at monthly timescales, we have now reproduced Figs. 3-5 and 8-10 with two changes:

i) higher line thickness for observed frequency and burned area,

ii) higher color transparency for the modeled uncertainty.

We hope the new plots allow the results from SMLFire1.0 to be compared more easily with observations.

We would like to thank all the reviewers for their detailed comments as well as critical feedback on our manuscript, and we look forward to a favorable response.

On behalf of the authors,

Jatan Buch